# Predicting and improving complex beer flavor through machine learning

Michiel Schreurs [1,2,3,7], Supinya Piampongsant[1,2,3,7], Miguel Roncoroni [1,2,3,7], Lloyd Cool [1,2,3,4], Beatriz Herrera-Malaver [1,2,3], Christophe Vanderaa [4], Florian A. Theßeling[1,2,3], Łukasz Kreft [5], Alexander Botzki [5], Philippe Malcorps[6], Luk Daenen[6], Tom Wenseleers [4] & Kevin J. Verstrepen [1,2,3] ✉

The perception and appreciation of food flavor depends on many interacting chemical compounds and external factors, and therefore proves challenging to understand and predict. Here, we combine extensive chemical and sensory analyses of 250 different beers to train machine learning models that allow predicting flavor and consumer appreciation. For each beer, we measure over 200 chemical properties, perform quantitative descriptive sensory analysis with a trained tasting panel and map data from over 180,000 consumer reviews to train 10 different machine learning models. The best-performing algorithm, Gradient Boosting, yields models that significantly outperform predictions based on conventional statistics and accurately predict complex food features and consumer appreciation from chemical profiles. Model dissection allows identifying specific and unexpected compounds as drivers of beer flavor and appreciation. Adding these compounds results in variants of commercial alcoholic and non-alcoholic beers with improved consumer appreciation. Together, our study reveals how big data and machine learning uncover complex links between food chemistry, flavor and consumer perception, and lays the foundation to develop novel, tailored foods with superior flavors.

Predicting and understanding food perception and appreciation is one of the major challenges in food science. Accurate modeling of food flavor and appreciation could yield important opportunities for both producers and consumers, including quality control, product fingerprinting, counterfeit detection, spoilage detection, and the development of new products and product combinations (food pairing)[1–6]. Accurate models for flavor and consumer appreciation would contribute greatly to our scientific understanding of how humans perceive and appreciate flavor. Moreover, accurate predictive models would also facilitate and standardize existing food assessment methods and could supplement or replace assessments by trained and consumer tasting panels, which are variable, expensive and time-consuming[7–9]. Lastly, apart from providing objective, quantitative, accurate and contextual information that can help producers, models can also guide consumers in understanding their personal preferences[10].

Despite the myriad of applications, predicting food flavor and appreciation from its chemical properties remains a largely elusive goal in sensory science, especially for complex food and beverages[11,12].

[1]VIB—KU Leuven Center for Microbiology, Gaston Geenslaan 1, B-3001 Leuven, Belgium. [2]CMPG Laboratory of Genetics and Genomics, KU Leuven, Gaston Geenslaan 1, B-3001 Leuven, Belgium. [3]Leuven Institute for Beer Research (LIBR), Gaston Geenslaan 1, B-3001 Leuven, Belgium. [4]Laboratory of Socioecology and Social Evolution, KU Leuven, Naamsestraat 59, B-3000 Leuven, Belgium. [5]VIB Bioinformatics Core, VIB, Rijvisschestraat 120, B-9052 Ghent, Belgium. [6]AB InBev SA/NV, Brouwerijplein 1, B-3000 Leuven, Belgium. [7]These authors contributed equally: Michiel Schreurs, Supinya Piampongsant, Miguel Roncoroni. ✉e-mail: kevin.verstrepen@kuleuven.be

A key obstacle is the immense number of flavor-active chemicals underlying food flavor. Flavor compounds can vary widely in chemical structure and concentration, making them technically challenging and labor-intensive to quantify, even in the face of innovations in metabolomics, such as non-targeted metabolic fingerprinting[13,14]. Moreover, sensory analysis is perhaps even more complicated. Flavor perception is highly complex, resulting from hundreds of different molecules interacting at the physiochemical and sensorial level. Sensory perception is often non-linear, characterized by complex and concentration-dependent synergistic and antagonistic effects[15–21] that are further convoluted by the genetics, environment, culture and psychology of consumers[22–24]. Perceived flavor is therefore difficult to measure, with problems of sensitivity, accuracy, and reproducibility that can only be resolved by gathering sufficiently large datasets[25]. Trained tasting panels are considered the prime source of quality sensory data, but require meticulous training, are low throughput and high cost. Public databases containing consumer reviews of food products could provide a valuable alternative, especially for studying appreciation scores, which do not require formal training[25]. Public databases offer the advantage of amassing large amounts of data, increasing the statistical power to identify potential drivers of appreciation. However, public datasets suffer from biases, including a bias in the volunteers that contribute to the database, as well as confounding factors such as price, cult status and psychological conformity towards previous ratings of the product.

Classical multivariate statistics and machine learning methods have been used to predict flavor of specific compounds by, for example, linking structural properties of a compound to its potential biological activities or linking concentrations of specific compounds to sensory profiles[1,26]. Importantly, most previous studies focused on predicting organoleptic properties of single compounds (often based on their chemical structure)[27–33], thus ignoring the fact that these compounds are present in a complex matrix in food or beverages and excluding complex interactions between compounds. Moreover, the classical statistics commonly used in sensory science[34–39] require a large sample size and sufficient variance amongst predictors to create accurate models. They are not fit for studying an extensive set of hundreds of interacting flavor compounds, since they are sensitive to outliers, have a high tendency to overfit and are less suited for non-linear and discontinuous relationships[40].

In this study, we combine extensive chemical analyses and sensory data of a set of different commercial beers with machine learning approaches to develop models that predict taste, smell, mouthfeel and appreciation from compound concentrations. Beer is particularly suited to model the relationship between chemistry, flavor and appreciation. First, beer is a complex product, consisting of thousands of flavor compounds that partake in complex sensory interactions[41–43]. This chemical diversity arises from the raw materials (malt, yeast, hops, water and spices) and biochemical conversions during the brewing process (kilning, mashing, boiling, fermentation, maturation and aging)[44,45]. Second, the advent of the internet saw beer consumers embrace online review platforms, such as RateBeer (ZX Ventures, Anheuser-Busch InBev SA/NV) and BeerAdvocate (Next Glass, inc.). In this way, the beer community provides massive data sets of beer flavor and appreciation scores, creating extraordinarily large sensory databases to complement the analyses of our professional sensory panel. Specifically, we characterize over 200 chemical properties of 250 commercial beers, spread across 22 beer styles, and link these to the descriptive sensory profiling data of a 16-person in-house trained tasting panel and data acquired from over 180,000 public consumer reviews. These unique and extensive datasets enable us to train a suite of machine learning models to predict flavor and appreciation from a beer's chemical profile. Dissection of the best-performing models allows us to pinpoint specific compounds as potential drivers of beer flavor and appreciation. Follow-up experiments confirm the importance of these compounds and ultimately allow us to significantly improve the flavor and appreciation of selected commercial beers. Together, our study represents a significant step towards understanding complex flavors and reinforces the value of machine learning to develop and refine complex foods. In this way, it represents a stepping stone for further computer-aided food engineering applications[46].

## Results

To generate a comprehensive dataset on beer flavor, we selected 250 commercial Belgian beers across 22 different beer styles (Supplementary Fig. S1). Beers with ≤ 4.2% alcohol by volume (ABV) were classified as non-alcoholic and low-alcoholic. Blonds and Tripels constitute a significant portion of the dataset (12.4% and 11.2%, respectively) reflecting their presence on the Belgian beer market and the heterogeneity of beers within these styles. By contrast, lager beers are less diverse and dominated by a handful of brands. Rare styles such as Brut or Faro make up only a small fraction of the dataset (2% and 1%, respectively) because fewer of these beers are produced and because they are dominated by distinct characteristics in terms of flavor and chemical composition.

### Extensive analysis identifies relationships between chemical compounds in beer

For each beer, we measured 226 different chemical properties, including common brewing parameters such as alcohol content, iso-alpha acids, pH, sugar concentration[47], and over 200 flavor compounds (Methods, Supplementary Table S1). A large portion (37.2%) are terpenoids arising from hopping, responsible for herbal and fruity flavors[16,48]. A second major category are yeast metabolites, such as esters and alcohols, that result in fruity and solvent notes[48–50]. Other measured compounds are primarily derived from malt, or other microbes such as non-*Saccharomyces* yeasts and bacteria ('wild flora'). Compounds that arise from spices or staling are labeled under 'Others'. Five attributes (caloric value, total acids and total ester, hop aroma and sulfur compounds) are calculated from multiple individually measured compounds.

As a first step in identifying relationships between chemical properties, we determined correlations between the concentrations of the compounds (Fig. 1, upper panel, Supplementary Data 1 and 2, and Supplementary Fig. S2. For the sake of clarity, only a subset of the measured compounds is shown in Fig. 1). Compounds of the same origin typically show a positive correlation, while absence of correlation hints at parameters varying independently. For example, the hop aroma compounds citronellol, and alpha-terpineol show moderate correlations with each other (Spearman's rho=0.39 and 0.57), but not with the bittering hop component iso-alpha acids (Spearman's rho=0.16 and −0.07). This illustrates how brewers can independently modify hop aroma and bitterness by selecting hop varieties and dosage time. If hops are added early in the boiling phase, chemical conversions increase bitterness while aromas evaporate, conversely, late addition of hops preserves aroma but limits bitterness[51]. Similarly, hop-derived iso-alpha acids show a strong anti-correlation with lactic acid and acetic acid, likely reflecting growth inhibition of lactic acid and acetic acid bacteria, or the consequent use of fewer hops in sour beer styles, such as West Flanders ales and Fruit beers, that rely on these bacteria for their distinct flavors[52]. Finally, yeast-derived esters (ethyl acetate, ethyl decanoate, ethyl hexanoate, ethyl octanoate) and alcohols (ethanol, isoamyl alcohol, isobutanol, and glycerol), correlate with Spearman coefficients above 0.5, suggesting that these secondary metabolites are correlated with the yeast genetic background and/or fermentation parameters and may be difficult to influence individually, although the choice of yeast strain may offer some control[53].

Interestingly, different beer styles show distinct patterns for some flavor compounds (Supplementary Fig. S3). These observations

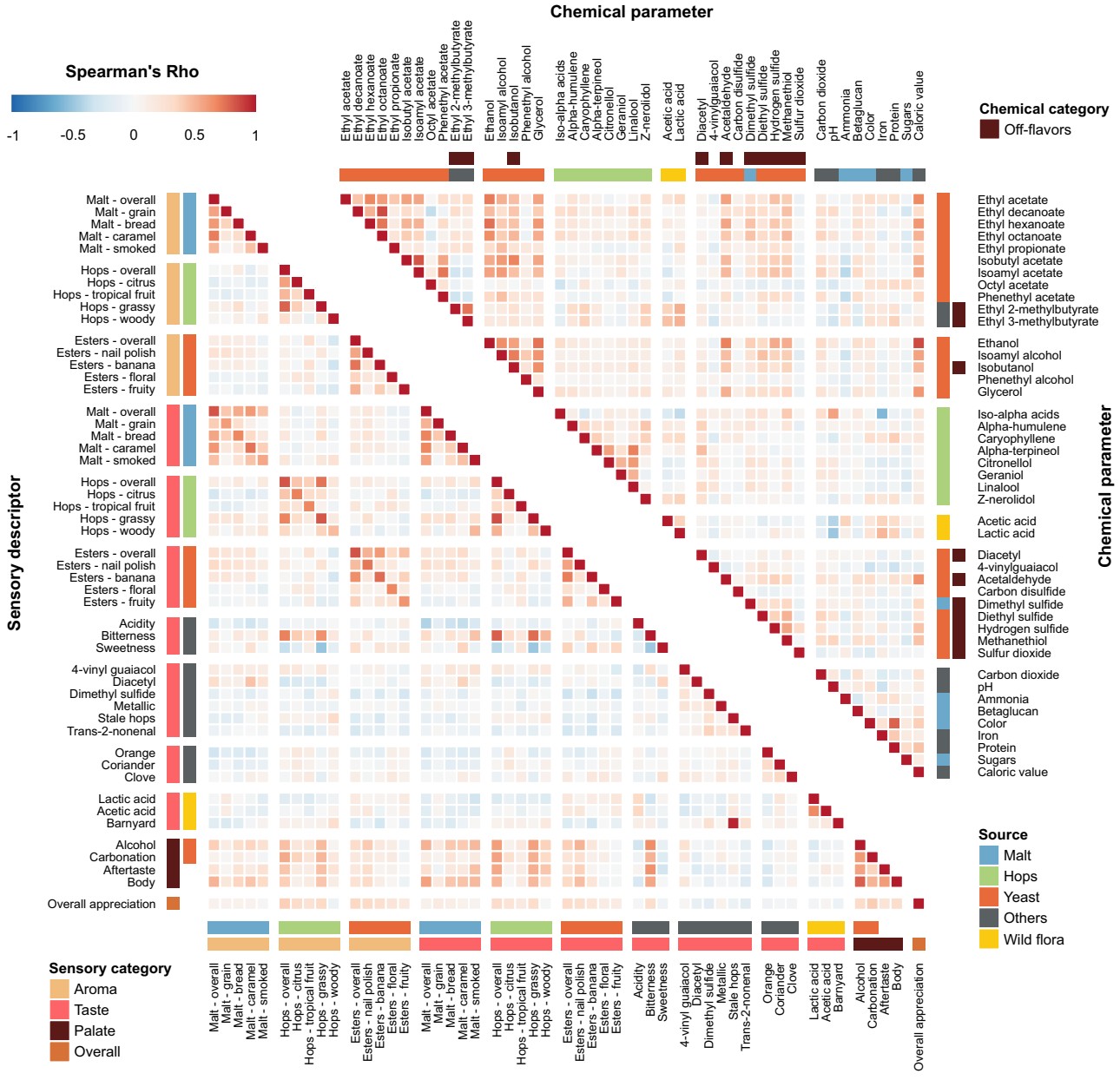

**Fig. 1 | Correlations between selected chemical parameters (upper right panel) and sensory descriptors used by the tasting panel (bottom left panel).** Spearman rank correlations are shown. Descriptors are grouped according to their origin (malt (blue), hops (green), yeast (red), wild flora (yellow), Others (black)), and sensory aspect (aroma, taste, palate, and overall appreciation). Please note that for the chemical compounds, for the sake of clarity, only a subset of the total number of measured compounds is shown, with an emphasis on the key compounds for each source. For more details, see the main text and Methods section. Chemical data can be found in Supplementary Data 1, correlations between all chemical compounds are depicted in Supplementary Fig. S2 and correlation values can be found in Supplementary Data 2. See Supplementary Data 4 for sensory panel assessments and Supplementary Data 5 for correlation values between all sensory descriptors.

agree with expectations for key beer styles, and serve as a control for our measurements. For instance, Stouts generally show high values for color (darker), while hoppy beers contain elevated levels of iso-alpha acids, compounds associated with bitter hop taste. Acetic and lactic acid are not prevalent in most beers, with notable exceptions such as Kriek, Lambic, Faro, West Flanders ales and Flanders Old Brown, which use acid-producing bacteria (*Lactobacillus* and *Pediococcus*) or unconventional yeast (*Brettanomyces*)[54,55]. Glycerol, ethanol and esters show similar distributions across all beer styles, reflecting their common origin as products of yeast metabolism during fermentation[45,53]. Finally, low/no-alcohol beers contain low

concentrations of glycerol and esters. This is in line with the production process for most of the low/no-alcohol beers in our dataset, which are produced through limiting fermentation or by stripping away alcohol via evaporation or dialysis, with both methods having the unintended side-effect of reducing the amount of flavor compounds in the final beer[56,57].

Besides expected associations, our data also reveals less trivial associations between beer styles and specific parameters. For example, geraniol and citronellol, two monoterpenoids responsible for citrus, floral and rose flavors and characteristic of Citra hops, are found in relatively high amounts in Christmas, Saison, and Brett/co-fermented

beers, where they may originate from terpenoid-rich spices such as coriander seeds instead of hops[58].

## Tasting panel assessments reveal sensorial relationships in beer

To assess the sensory profile of each beer, a trained tasting panel evaluated each of the 250 beers for 50 sensory attributes, including different hop, malt and yeast flavors, off-flavors and spices. Panelists used a tasting sheet (Supplementary Data 3) to score the different attributes. Panel consistency was evaluated by repeating 12 samples across different sessions and performing ANOVA. In 95% of cases no significant difference was found across sessions ($p > 0.05$), indicating good panel consistency (Supplementary Table S2).

Aroma and taste perception reported by the trained panel are often linked (Fig. 1, bottom left panel and Supplementary Data 4 and 5), with high correlations between hops aroma and taste (Spearman's rho=0.83). Bitter taste was found to correlate with hop aroma and taste in general (Spearman's rho=0.80 and 0.69), and particularly with "grassy" noble hops (Spearman's rho=0.75). Barnyard flavor, most often associated with sour beers, is identified together with stale hops (Spearman's rho=0.97) that are used in these beers. Lactic and acetic acid, which often co-occur, are correlated (Spearman's rho=0.66). Interestingly, sweetness and bitterness are anti-correlated (Spearman's rho = −0.48), confirming the hypothesis that they mask each other[59,60]. Beer body is highly correlated with alcohol (Spearman's rho = 0.79), and overall appreciation is found to correlate with multiple aspects that describe beer mouthfeel (alcohol, carbonation; Spearman's rho= 0.32, 0.39), as well as with hop and ester aroma intensity (Spearman's rho=0.39 and 0.35).

Similar to the chemical analyses, sensorial analyses confirmed typical features of specific beer styles (Supplementary Fig. S4). For example, sour beers (Faro, Flanders Old Brown, Fruit beer, Kriek, Lambic, West Flanders ale) were rated acidic, with flavors of both acetic and lactic acid. Hoppy beers were found to be bitter and showed hop-associated aromas like citrus and tropical fruit. Malt taste is most detected among scotch, stout/porters, and strong ales, while low/no-alcohol beers, which often have a reputation for being 'worty' (reminiscent of unfermented, sweet malt extract) appear in the middle. Unsurprisingly, hop aromas are most strongly detected among hoppy beers. Like its chemical counterpart (Supplementary Fig. S3), acidity shows a right-skewed distribution, with the most acidic beers being Krieks, Lambics, and West Flanders ales.

## Tasting panel assessments of specific flavors correlate with chemical composition

We find that the concentrations of several chemical compounds strongly correlate with specific aroma or taste, as evaluated by the tasting panel (Fig. 2, Supplementary Fig. S5, Supplementary Data 6). In some cases, these correlations confirm expectations and serve as a useful control for data quality. For example, iso-alpha acids, the bittering compounds in hops, strongly correlate with bitterness (Spearman's rho=0.68), while ethanol and glycerol correlate with tasters' perceptions of alcohol and body, the mouthfeel sensation of fullness (Spearman's rho=0.82/0.62 and 0.72/0.57 respectively) and darker color from roasted malts is a good indication of malt perception (Spearman's rho=0.54).

Interestingly, for some relationships between chemical compounds and perceived flavor, correlations are weaker than expected. For example, the rose-smelling phenethyl acetate only weakly correlates with floral aroma. This hints at more complex relationships and interactions between compounds and suggests a need for a more complex model than simple correlations. Lastly, we uncovered unexpected correlations. For instance, the esters ethyl decanoate and ethyl octanoate appear to correlate slightly with hop perception and bitterness, possibly due to their fruity flavor. Iron is anti-correlated with hop aromas and bitterness, most likely because it is also anti-

correlated with iso-alpha acids. This could be a sign of metal chelation of hop acids[61], given that our analyses measure unbound hop acids and total iron content, or could result from the higher iron content in dark and Fruit beers, which typically have less hoppy and bitter flavors[62].

## Public consumer reviews complement expert panel data

To complement and expand the sensory data of our trained tasting panel, we collected 180,000 reviews of our 250 beers from the online consumer review platform RateBeer. This provided numerical scores for beer appearance, aroma, taste, palate, overall quality as well as the average overall score.

Public datasets are known to suffer from biases, such as price, cult status and psychological conformity towards previous ratings of a product. For example, prices correlate with appreciation scores for these online consumer reviews (rho=0.49, Supplementary Fig. S6), but not for our trained tasting panel (rho=0.19). This suggests that prices affect consumer appreciation, which has been reported in wine[63], while blind tastings are unaffected. Moreover, we observe that some beer styles, like lagers and non-alcoholic beers, generally receive lower scores, reflecting that online reviewers are mostly beer aficionados with a preference for specialty beers over lager beers. In general, we find a modest correlation between our trained panel's overall appreciation score and the online consumer appreciation scores (Fig. 3, rho=0.29). Apart from the aforementioned biases in the online datasets, serving temperature, sample freshness and surroundings, which are all tightly controlled during the tasting panel sessions, can vary tremendously across online consumers and can further contribute to (among others, appreciation) differences between the two categories of tasters. Importantly, in contrast to the overall appreciation scores, for many sensory aspects the results from the professional panel correlated well with results obtained from RateBeer reviews. Correlations were highest for features that are relatively easy to recognize even for untrained tasters, like bitterness, sweetness, alcohol and malt aroma (Fig. 3 and below).

Besides collecting consumer appreciation from these online reviews, we developed automated text analysis tools to gather additional data from review texts (Supplementary Data 7). Processing review texts on the RateBeer database yielded comparable results to the scores given by the trained panel for many common sensory aspects, including acidity, bitterness, sweetness, alcohol, malt, and hop tastes (Fig. 3). This is in line with what would be expected, since these attributes require less training for accurate assessment and are less influenced by environmental factors such as temperature, serving glass and odors in the environment. Consumer reviews also correlate well with our trained panel for 4-vinyl guaiacol, a compound associated with a very characteristic aroma. By contrast, correlations for more specific aromas like ester, coriander or diacetyl are underrepresented in the online reviews, underscoring the importance of using a trained tasting panel and standardized tasting sheets with explicit factors to be scored for evaluating specific aspects of a beer. Taken together, our results suggest that public reviews are trustworthy for some, but not all, flavor features and can complement or substitute taste panel data for these sensory aspects.

## Models can predict beer sensory profiles from chemical data

The rich datasets of chemical analyses, tasting panel assessments and public reviews gathered in the first part of this study provided us with a unique opportunity to develop predictive models that link chemical data to sensorial features. Given the complexity of beer flavor, basic statistical tools such as correlations or linear regression may not always be the most suitable for making accurate predictions. Instead, we applied different machine learning models that can model both simple linear and complex interactive relationships. Specifically, we constructed a set of regression models to predict (a) trained panel

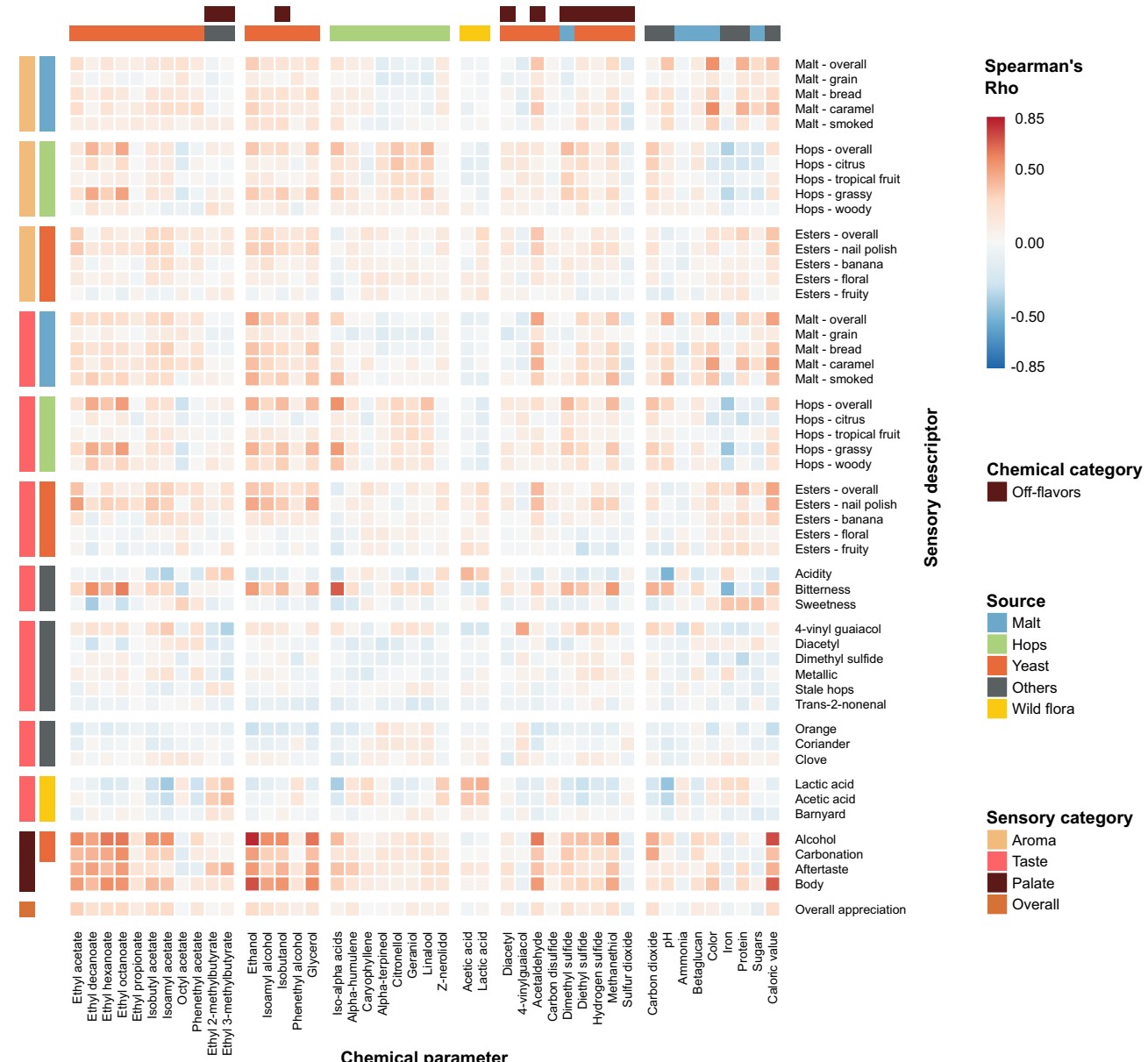

**Fig. 2 | Pairwise Spearman Rank correlations between chemical data and sensorial data from the trained panel.** Heatmap colors indicate Spearman's Rho. Axes are organized according to sensory categories (aroma, taste, mouthfeel, overall), chemical categories and chemical sources in beer (malt (blue), hops (green), yeast (red), wild flora (yellow), Others (black)). See Supplementary Data 6 for all correlation values.

scores for beer flavor and quality and (b) public reviews' appreciation scores from beer chemical profiles. We trained and tested 10 different models (Methods), 3 linear regression-based models (simple linear regression with first-order interactions (LR), lasso regression with first-order interactions (Lasso), partial least squares regressor (PLSR)), 5 decision tree models (AdaBoost regressor (ABR), extra trees (ET), gradient boosting regressor (GBR), random forest (RF) and XGBoost regressor (XGBR)), 1 support vector regression (SVR), and 1 artificial neural network (ANN) model.

To compare the performance of our machine learning models, the dataset was randomly split into a training and test set, stratified by beer style. After a model was trained on data in the training set, its performance was evaluated on its ability to predict the test dataset obtained from multi-output models (based on the coefficient of determination, see Methods). Additionally, individual-attribute models were ranked per descriptor and the average rank was calculated, as proposed by Korneva et al.[64]. Importantly, both ways of evaluating the models'

performance agreed in general. Performance of the different models varied (Table 1). It should be noted that all models perform better at predicting RateBeer results than results from our trained tasting panel. One reason could be that sensory data is inherently variable, and this variability is averaged out with the large number of public reviews from RateBeer. Additionally, all tree-based models perform better at predicting taste than aroma. Linear models (LR) performed particularly poorly, with negative $R^2$ values, due to severe overfitting (training set $R^2 = 1$). Overfitting is a common issue in linear models with many parameters and limited samples, especially with interaction terms further amplifying the number of parameters. L1 regularization (Lasso) successfully overcomes this overfitting, out-competing multiple tree-based models on the RateBeer dataset. Similarly, the dimensionality reduction of PLSR avoids overfitting and improves performance, to some extent. Still, tree-based models (ABR, ET, GBR, RF and XGBR) show the best performance, out-competing the linear models (LR, Lasso, PLSR) commonly used in sensory science[65].

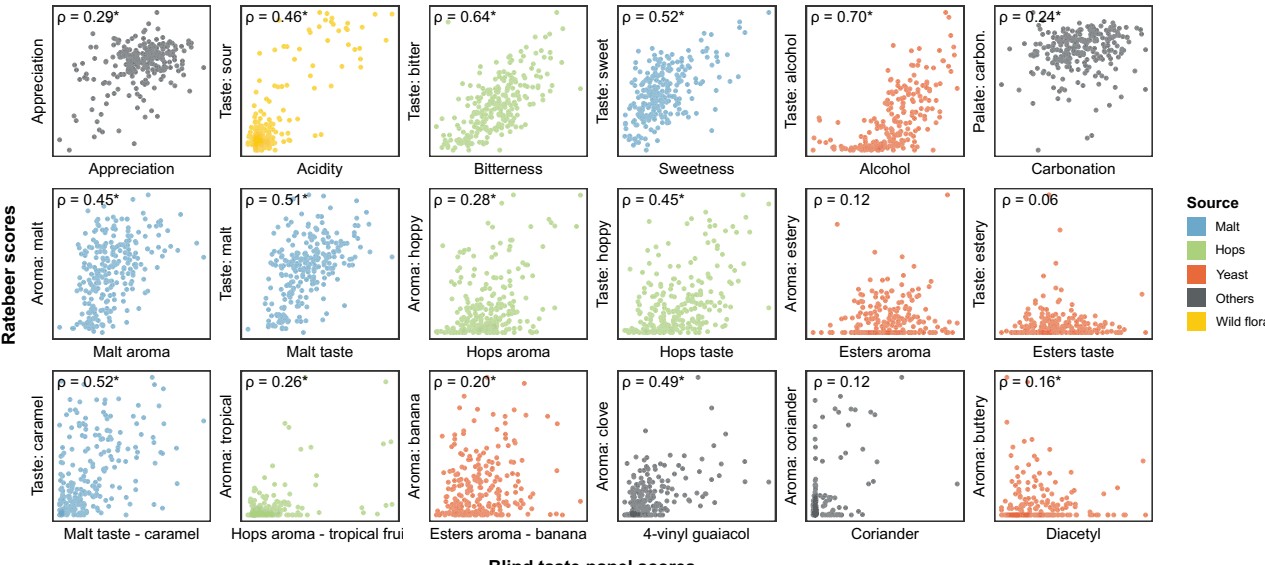

**Fig. 3 | Correlations between online reviews and trained tasting panel scores.** RateBeer text mining results can be found in Supplementary Data 7. Rho values shown are Spearman correlation values, with asterisks indicating significant correlations ($p < 0.05$, two-sided). All $p$ values were smaller than 0.001, except for Esters aroma (0.0553), Esters taste (0.3275), Esters aroma−banana (0.0019), Coriander (0.0508) and Diacetyl (0.0134).

**Table 1 | Performance of various models when trained to predict either trained tasting panel descriptors or RateBeer review scores**

| Model | Trained panel $R^2$ | Trained panel rank | RateBeer $R^2$ | RateBeer rank |
|---|---|---|---|---|
| AdaBoost | 0.21 | 2.88 | 0.61 | 5.67 |
| Artificial Neural Network | 0.14 | 5.60 | 0.46 | 4.00 |
| Extra Trees | **0.22** | 3.02 | 0.61 | 4.67 |
| Gradient boosting | 0.21 | 3.42 | **0.69** | **1.50** |
| Lasso regression | 0.05 | 4.94 | 0.64 | 4.33 |
| Linear regression | −4.13 | 7.88 | −11.02 | 8.00 |
| Partial Least Squares Regression | −0.25 | 7.56 | 0.57 | 5.33 |
| Random Forest | **0.22** | **2.86** | 0.62 | 3.50 |
| Support Vector Regression | 0.18 | 6.50 | 0.59 | 6.50 |
| XGBoost | **0.22** | 4.12 | 0.62 | 2.00 |

The performance metric is the coefficient of determination ($R^2$) for predictions on the test dataset, obtained from multi-output models (Methods). The average rank is the mean after ranking the individual-attribute models per descriptor, with the lowest value indicating the best model. The highest scores per evaluation metric are indicated in bold. Note that some models result in negative R-squared values, implying the average of the outcome variable would have worked as a better predictor than the models' predictions. Values for all descriptors can be found in Supplementary Table S3 (tasting panel) and Supplementary Table S4 (RateBeer).

GBR models showed the best overall performance in predicting sensory responses from chemical information, with $R^2$ values up to 0.75 depending on the predicted sensory feature (Supplementary Table S4). The GBR models predict consumer appreciation (RateBeer) better than our trained panel's appreciation ($R^2$ value of 0.67 compared to $R^2$ value of 0.09) (Supplementary Table S3 and Supplementary Table S4). ANN models showed intermediate performance, likely because neural networks typically perform best with larger datasets[66]. The SVR shows intermediate performance, mostly due to the weak predictions of specific attributes that lower the overall performance (Supplementary Table S4).

**Model dissection identifies specific, unexpected compounds as drivers of consumer appreciation**

Next, we leveraged our models to infer important contributors to sensory perception and consumer appreciation. Consumer preference is a crucial sensory aspects, because a product that shows low consumer appreciation scores often does not succeed commercially[25].

Additionally, the requirement for a large number of representative evaluators makes consumer trials one of the more costly and time-consuming aspects of product development. Hence, a model for predicting chemical drivers of overall appreciation would be a welcome addition to the available toolbox for food development and optimization.

Since GBR models on our RateBeer dataset showed the best overall performance, we focused on these models. Specifically, we used two approaches to identify important contributors. First, rankings of the most important predictors for each sensorial trait in the GBR models were obtained based on impurity-based feature importance (mean decrease in impurity). High-ranked parameters were hypothesized to be either the true causal chemical properties underlying the trait, to correlate with the actual causal properties, or to take part in sensory interactions affecting the trait[67] (Fig. 4A). In a second approach, we used SHAP[68] to determine which parameters contributed most to the model for making predictions of consumer appreciation (Fig. 4B). SHAP calculates parameter contributions to model

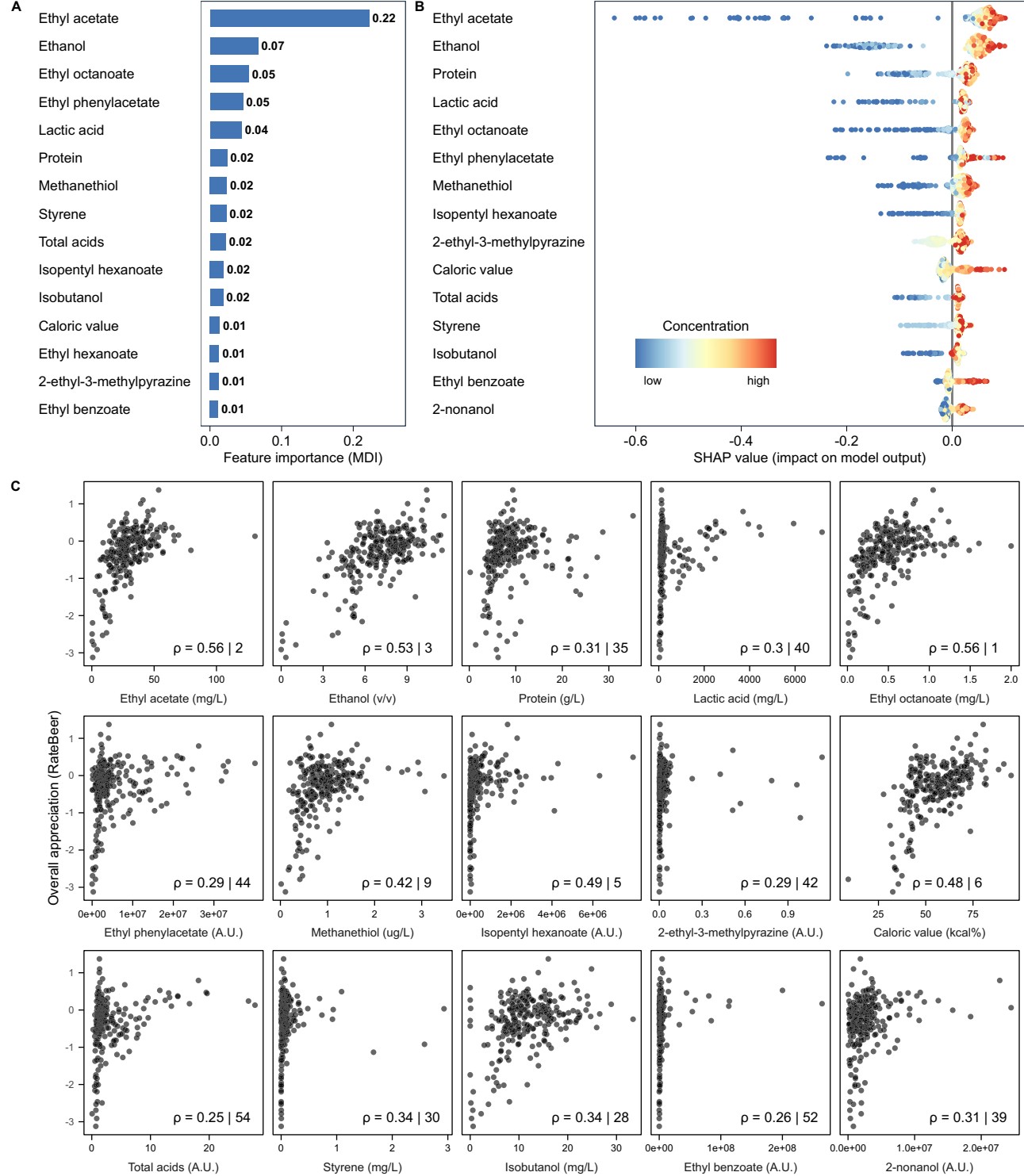

**Fig. 4 | Most important chemical parameters. A** The impurity-based feature importance (mean deviance in impurity, MDI) calculated from the Gradient Boosting Regression (GBR) model predicting RateBeer appreciation scores. The top 15 highest ranked chemical properties are shown. **B** SHAP summary plot for the top 15 parameters contributing to our GBR model. Each point on the graph represents a sample from our dataset. The color represents the concentration of that parameter, with bluer colors representing low values and redder colors representing higher values. Greater absolute values on the horizontal axis indicate a higher impact of the parameter on the prediction of the model. **C** Spearman correlations between the 15 most important chemical properties and consumer overall appreciation. Numbers indicate the Spearman Rho correlation coefficient, and the rank of this correlation compared to all other correlations. The top 15 important compounds were determined using SHAP (panel B).

predictions on a per-sample basis, which can be aggregated into an importance score.

Both approaches identified ethyl acetate as the most predictive parameter for beer appreciation (Fig. 4). Ethyl acetate is the most abundant ester in beer with a typical 'fruity', 'solvent' and 'alcoholic' flavor, but is often considered less important than other esters like isoamyl acetate. The second most important parameter identified by SHAP is ethanol, the most abundant beer compound after water.

Apart from directly contributing to beer flavor and mouthfeel, ethanol drastically influences the physical properties of beer, dictating how easily volatile compounds escape the beer matrix to contribute to beer aroma[69]. Importantly, it should also be noted that the importance of ethanol for appreciation is likely inflated by the very low appreciation scores of non-alcoholic beers (Supplementary Fig. S4). Despite not often being considered a driver of beer appreciation, protein level also ranks highly in both approaches, possibly due to its effect on mouthfeel and body[70]. Lactic acid, which contributes to the tart taste of sour beers, is the fourth most important parameter identified by SHAP, possibly due to the generally high appreciation of sour beers in our dataset.

Interestingly, some of the most important predictive parameters for our model are not well-established as beer flavors or are even commonly regarded as being negative for beer quality. For example, our models identify methanethiol and ethyl phenyl acetate, an ester commonly linked to beer staling[71], as a key factor contributing to beer appreciation. Although there is no doubt that high concentrations of these compounds are considered unpleasant, the positive effects of modest concentrations are not yet known[72,73].

To compare our approach to conventional statistics, we evaluated how well the 15 most important SHAP-derived parameters correlate with consumer appreciation (Fig. 4C). Interestingly, only 6 of the properties derived by SHAP rank amongst the top 15 most correlated parameters. For some chemical compounds, the correlations are so low that they would have likely been considered unimportant. For example, lactic acid, the fourth most important parameter, shows a bimodal distribution for appreciation, with sour beers forming a separate cluster, that is missed entirely by the Spearman correlation. Additionally, the correlation plots reveal outliers, emphasizing the need for robust analysis tools. Together, this highlights the need for alternative models, like the Gradient Boosting model, that better grasp the complexity of (beer) flavor.

Finally, to observe the relationships between these chemical properties and their predicted targets, partial dependence plots were constructed for the six most important predictors of consumer appreciation[74–76] (Supplementary Fig. S7). One-way partial dependence plots show how a change in concentration affects the predicted appreciation. These plots reveal an important limitation of our models: appreciation predictions remain constant at ever-increasing concentrations. This implies that once a threshold concentration is reached, further increasing the concentration does not affect appreciation. This is false, as it is well-documented that certain compounds become unpleasant at high concentrations, including ethyl acetate ('nail polish')[77] and methanethiol ('sulfury' and 'rotten cabbage')[78]. The inability of our models to grasp that flavor compounds have optimal levels, above which they become negative, is a consequence of working with commercial beer brands where (off-)flavors are rarely too high to negatively impact the product. The two-way partial dependence plots show how changing the concentration of two compounds influences predicted appreciation, visualizing their interactions (Supplementary Fig. S7). In our case, the top 5 parameters are dominated by additive or synergistic interactions, with high concentrations for both compounds resulting in the highest predicted appreciation.

To assess the robustness of our best-performing models and model predictions, we performed 100 iterations of the GBR, RF and ET models. In general, all iterations of the models yielded similar performance (Supplementary Fig. S8). Moreover, the main predictors (including the top predictors ethanol and ethyl acetate) remained virtually the same, especially for GBR and RF. For the iterations of the ET model, we did observe more variation in the top predictors, which is likely a consequence of the model's inherent random architecture in combination with co-correlations between certain predictors. However, even in this case, several of the top predictors (ethanol and ethyl

acetate) remain unchanged, although their rank in importance changes (Supplementary Fig. S8).

Next, we investigated if a combination of RateBeer and trained panel data into one consolidated dataset would lead to stronger models, under the hypothesis that such a model would suffer less from bias in the datasets. A GBR model was trained to predict appreciation on the combined dataset. This model underperformed compared to the RateBeer model, both in the native case and when including a dataset identifier ($R^2 = 0.67$, 0.26 and 0.42 respectively). For the latter, the dataset identifier is the most important feature (Supplementary Fig. S9), while most of the feature importance remains unchanged, with ethyl acetate and ethanol ranking highest, like in the original model trained only on RateBeer data. It seems that the large variation in the panel dataset introduces noise, weakening the models' performances and reliability. In addition, it seems reasonable to assume that both datasets are fundamentally different, with the panel dataset obtained by blind tastings by a trained professional panel.

Lastly, we evaluated whether beer style identifiers would further enhance the model's performance. A GBR model was trained with parameters that explicitly encoded the styles of the samples. This did not improve model performance ($R2 = 0.66$ with style information vs $R2 = 0.67$). The most important chemical features are consistent with the model trained without style information (eg. ethanol and ethyl acetate), and with the exception of the most preferred (strong ale) and least preferred (low/no-alcohol) styles, none of the styles were among the most important features (Supplementary Fig. S9, Supplementary Table S5 and S6). This is likely due to a combination of style-specific chemical signatures, such as iso-alpha acids and lactic acid, that implicitly convey style information to the original models, as well as the low number of samples belonging to some styles, making it difficult for the model to learn style-specific patterns. Moreover, beer styles are not rigorously defined, with some styles overlapping in features and some beers being misattributed to a specific style, all of which leads to more noise in models that use style parameters.

## Model validation

To test if our predictive models give insight into beer appreciation, we set up experiments aimed at improving existing commercial beers. We specifically selected overall appreciation as the trait to be examined because of its complexity and commercial relevance. Beer flavor comprises a complex bouquet rather than single aromas and tastes[53]. Hence, adding a single compound to the extent that a difference is noticeable may lead to an unbalanced, artificial flavor. Therefore, we evaluated the effect of combinations of compounds. Because Blond beers represent the most extensive style in our dataset, we selected a beer from this style as the starting material for these experiments (Beer 64 in Supplementary Data 1).

In the first set of experiments, we adjusted the concentrations of compounds that made up the most important predictors of overall appreciation (ethyl acetate, ethanol, lactic acid, ethyl phenyl acetate) together with correlated compounds (ethyl hexanoate, isoamyl acetate, glycerol), bringing them up to 95th percentile ethanol-normalized concentrations (Methods) within the Blond group ('Spiked' concentration in Fig. 5A). Compared to controls, the spiked beers were found to have significantly improved overall appreciation among trained panelists, with panelist noting increased intensity of ester flavors, sweetness, alcohol, and body fullness (Fig. 5B). To disentangle the contribution of ethanol to these results, a second experiment was performed without the addition of ethanol. This resulted in a similar outcome, including increased perception of alcohol and overall appreciation.

In a last experiment, we tested whether using the model's predictions can boost the appreciation of a non-alcoholic beer (beer 223 in Supplementary Data 1). Again, the addition of a mixture of predicted

**A**

**Blond: Best predictors**

| Compound | Control | Spiked | Threshold |
|---|---|---|---|
| Ethyl acetate (mg/l) | 15.59 | 33.12 | 23.00 |
| Ethyl hexanoate (mg/l) | 0.10 | 0.32 | 0.22 |
| Isoamyl acetate (mg/l) | 2.59 | 3.46 | 0.60 |
| Phenylethyl acetate (mg/l) | 0.36 | 0.39 | NA |
| Ethanol (v/v%) | 4.71 | 8.59 | 1.74 |
| Glycerol (g/l) | 1.51 | 3.21 | 2.00 |
| Lactic acid (mg/l) | 148.00 | 194.00 | 400.00 |

**Blond: Best predictors - no ethanol**

| Compound | Control | Spiked | Threshold |
|---|---|---|---|
| Ethyl acetate (mg/l) | 15.59 | 33.12 | 23.00 |
| Ethyl hexanoate (mg/l) | 0.10 | 0.32 | 0.22 |
| Isoamyl acetate (mg/l) | 2.59 | 3.46 | 0.60 |
| Phenylethyl acetate (mg/l) | 0.36 | 0.39 | NA |
| Glycerol (g/l) | 1.51 | 3.21 | 2.00 |
| Lactic acid (mg/l) | 148.00 | 194.00 | 400.00 |

**Non/low-alcohol: Best predictors - no ethanol**

| Compound | Control | Spiked | Threshold |
|---|---|---|---|
| Ethyl acetate (mg/l) | 4.27 | 8.43 | 23.00 |
| Ethyl hexanoate (mg/l) | 0.11 | 0.11 | 0.22 |
| Isoamyl acetate (mg/l) | 1.69 | 2.58 | 0.60 |
| Phenylethyl acetate (mg/l) | 0.07 | 0.33 | NA |
| Glycerol (g/l) | 1.63 | 17.59 | 2.00 |
| Lactic acid (mg/l) | 88.00 | 820.00 | 400.00 |

**B**

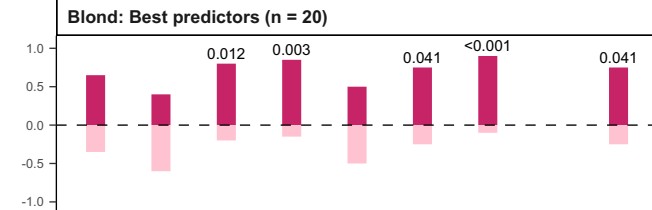

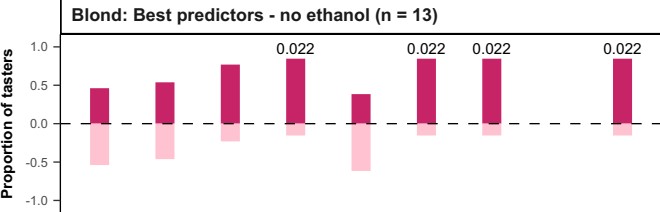

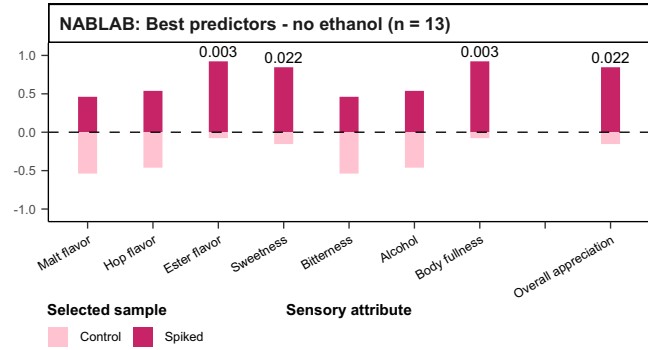

**Fig. 5 | Model validation by a beer supplemented with the top predicted chemical compounds.** Adding the top chemical compounds, identified as best predictors of appreciation by our model, into poorly appreciated beers results in increased appreciation from our trained panel. Results of sensory tests between base beers and those spiked with compounds identified as the best predictors by the model. **A** Blond and Non/Low-alcohol (0.0% ABV) base beers were brought up to 95th-percentile ethanol-normalized concentrations within each style. **B** For each sensory attribute, tasters indicated the more intense sample and selected the sample they preferred. The numbers above the bars correspond to the $p$ values that indicate significant changes in perceived flavor (two-sided binomial test: alpha 0.05, $n = 20$ or 13).

compounds (omitting ethanol, in this case) resulted in a significant increase in appreciation, body, ester flavor and sweetness.

## Discussion

Predicting flavor and consumer appreciation from chemical composition is one of the ultimate goals of sensory science. A reliable, systematic and unbiased way to link chemical profiles to flavor and food appreciation would be a significant asset to the food and beverage industry. Such tools would substantially aid in quality control and recipe development, offer an efficient and cost-effective alternative to pilot studies and consumer trials and would ultimately allow food manufacturers to produce superior, tailor-made products that better meet the demands of specific consumer groups more efficiently.

A limited set of studies have previously tried, to varying degrees of success, to predict beer flavor and beer popularity based on (a limited set of) chemical compounds and flavors[79,80]. Current sensitive, high-throughput technologies allow measuring an unprecedented number of chemical compounds and properties in a large set of samples, yielding a dataset that can train models that help close the gaps between chemistry and flavor, even for a complex natural product like beer. To our knowledge, no previous research gathered data at this scale (250 samples, 226 chemical parameters, 50 sensory attributes and 5 consumer scores) to disentangle and validate the chemical aspects driving beer preference using various machine-learning techniques. We find that modern machine learning models outperform conventional statistical tools, such as correlations and linear models,

and can successfully predict flavor appreciation from chemical composition. This could be attributed to the natural incorporation of interactions and non-linear or discontinuous effects in machine learning models, which are not easily grasped by the linear model architecture. While linear models and partial least squares regression represent the most widespread statistical approaches in sensory science, in part because they allow interpretation[65,81,82], modern machine learning methods allow for building better predictive models while preserving the possibility to dissect and exploit the underlying patterns. Of the 10 different models we trained, tree-based models, such as our best performing GBR, showed the best overall performance in predicting sensory responses from chemical information, outcompeting artificial neural networks. This agrees with previous reports for models trained on tabular data[83]. Our results are in line with the findings of Colantonio et al. who also identified the gradient boosting architecture as performing best at predicting appreciation and flavor (of tomatoes and blueberries, in their specific study)[26]. Importantly, besides our larger experimental scale, we were able to directly confirm our models' predictions in vivo.

Our study confirms that flavor compound concentration does not always correlate with perception, suggesting complex interactions that are often missed by more conventional statistics and simple models. Specifically, we find that tree-based algorithms may perform best in developing models that link complex food chemistry with aroma. Furthermore, we show that massive datasets of untrained consumer reviews provide a valuable source of data, that can

complement or even replace trained tasting panels, especially for appreciation and basic flavors, such as sweetness and bitterness. This holds despite biases that are known to occur in such datasets, such as price or conformity bias. Moreover, GBR models predict taste better than aroma. This is likely because taste (e.g. bitterness) often directly relates to the corresponding chemical measurements (e.g., iso-alpha acids), whereas such a link is less clear for aromas, which often result from the interplay between multiple volatile compounds. We also find that our models are best at predicting acidity and alcohol, likely because there is a direct relation between the measured chemical compounds (acids and ethanol) and the corresponding perceived sensorial attribute (acidity and alcohol), and because even untrained consumers are generally able to recognize these flavors and aromas.

The predictions of our final models, trained on review data, hold even for blind tastings with small groups of trained tasters, as demonstrated by our ability to validate specific compounds as drivers of beer flavor and appreciation. Since adding a single compound to the extent of a noticeable difference may result in an unbalanced flavor profile, we specifically tested our identified key drivers as a combination of compounds. While this approach does not allow us to validate if a particular single compound would affect flavor and/or appreciation, our experiments do show that this combination of compounds increases consumer appreciation.

It is important to stress that, while it represents an important step forward, our approach still has several major limitations. A key weakness of the GBR model architecture is that amongst co-correlating variables, the largest main effect is consistently preferred for model building. As a result, co-correlating variables often have artificially low importance scores, both for impurity and SHAP-based methods, like we observed in the comparison to the more randomized Extra Trees models. This implies that chemicals identified as key drivers of a specific sensory feature by GBR might not be the true causative compounds, but rather co-correlate with the actual causative chemical. For example, the high importance of ethyl acetate could be (partially) attributed to the total ester content, ethanol or ethyl hexanoate (rho=0.77, rho=0.72 and rho=0.68), while ethyl phenylacetate could hide the importance of prenyl isobutyrate and ethyl benzoate (rho=0.77 and rho=0.76). Expanding our GBR model to include beer style as a parameter did not yield additional power or insight. This is likely due to style-specific chemical signatures, such as iso-alpha acids and lactic acid, that implicitly convey style information to the original model, as well as the smaller sample size per style, limiting the power to uncover style-specific patterns. This can be partly attributed to the curse of dimensionality, where the high number of parameters results in the models mainly incorporating single parameter effects, rather than complex interactions such as style-dependent effects[67]. A larger number of samples may overcome some of these limitations and offer more insight into style-specific effects. On the other hand, beer style is not a rigid scientific classification, and beers within one style often differ a lot, which further complicates the analysis of style as a model factor.

Our study is limited to beers from Belgian breweries. Although these beers cover a large portion of the beer styles available globally, some beer styles and consumer patterns may be missing, while other features might be overrepresented. For example, many Belgian ales exhibit yeast-driven flavor profiles, which is reflected in the chemical drivers of appreciation discovered by this study. In future work, expanding the scope to include diverse markets and beer styles could lead to the identification of even more drivers of appreciation and better models for special niche products that were not present in our beer set.

In addition to inherent limitations of GBR models, there are also some limitations associated with studying food aroma. Even if our chemical analyses measured most of the known aroma compounds, the total number of flavor compounds in complex foods like beer is still larger than the subset we were able to measure in this study. For example, hop-derived thiols, that influence flavor at very low concentrations, are notoriously difficult to measure in a high-throughput experiment. Moreover, consumer perception remains subjective and prone to biases that are difficult to avoid. It is also important to stress that the models are still immature and that more extensive datasets will be crucial for developing more complete models in the future. Besides more samples and parameters, our dataset does not include any demographic information about the tasters. Including such data could lead to better models that grasp external factors like age and culture. Another limitation is that our set of beers consists of high-quality end-products and lacks beers that are unfit for sale, which limits the current model in accurately predicting products that are appreciated very badly. Finally, while models could be readily applied in quality control, their use in sensory science and product development is restrained by their inability to discern causal relationships. Given that the models cannot distinguish compounds that genuinely drive consumer perception from those that merely correlate, validation experiments are essential to identify true causative compounds.

Despite the inherent limitations, dissection of our models enabled us to pinpoint specific molecules as potential drivers of beer aroma and consumer appreciation, including compounds that were unexpected and would not have been identified using standard approaches. Important drivers of beer appreciation uncovered by our models include protein levels, ethyl acetate, ethyl phenyl acetate and lactic acid. Currently, many brewers already use lactic acid to acidify their brewing water and ensure optimal pH for enzymatic activity during the mashing process. Our results suggest that adding lactic acid can also improve beer appreciation, although its individual effect remains to be tested. Interestingly, ethanol appears to be unnecessary to improve beer appreciation, both for blond beer and alcohol-free beer. Given the growing consumer interest in alcohol-free beer, with a predicted annual market growth of >7%[84], it is relevant for brewers to know what compounds can further increase consumer appreciation of these beers. Hence, our model may readily provide avenues to further improve the flavor and consumer appreciation of both alcoholic and non-alcoholic beers, which is generally considered one of the key challenges for future beer production.

Whereas we see a direct implementation of our results for the development of superior alcohol-free beverages and other food products, our study can also serve as a stepping stone for the development of novel alcohol-containing beverages. We want to echo the growing body of scientific evidence for the negative effects of alcohol consumption, both on the individual level by the mutagenic, teratogenic and carcinogenic effects of ethanol[85,86], as well as the burden on society caused by alcohol abuse and addiction. We encourage the use of our results for the production of healthier, tastier products, including novel and improved beverages with lower alcohol contents. Furthermore, we strongly discourage the use of these technologies to improve the appreciation or addictive properties of harmful substances.

The present work demonstrates that despite some important remaining hurdles, combining the latest developments in chemical analyses, sensory analysis and modern machine learning methods offers exciting avenues for food chemistry and engineering. Soon, these tools may provide solutions in quality control and recipe development, as well as new approaches to sensory science and flavor research.

## Methods
### Beer selection
250 commercial Belgian beers were selected to cover the broad diversity of beer styles and corresponding diversity in chemical composition and aroma. See Supplementary Fig. S1.

## Chemical dataset

**Sample preparation.** Beers within their expiration date were purchased from commercial retailers. Samples were prepared in biological duplicates at room temperature, unless explicitly stated otherwise. Bottle pressure was measured with a manual pressure device (Steinfurth Mess-Systeme GmbH) and used to calculate $CO_2$ concentration. The beer was poured through two filter papers (Macherey-Nagel, 500713032 MN 713 ¼) to remove carbon dioxide and prevent spontaneous foaming. Samples were then prepared for measurements by targeted Headspace-Gas Chromatography-Flame Ionization Detector/Flame Photometric Detector (HS-GC-FID/FPD), Headspace-Solid Phase Microextraction-Gas Chromatography-Mass Spectrometry (HS-SPME-GC-MS), colorimetric analysis, enzymatic analysis, Near-Infrared (NIR) analysis, as described in the sections below. The mean values of biological duplicates are reported for each compound.

**HS-GC-FID/FPD.** HS-GC-FID/FPD (Shimadzu GC 2010 Plus) was used to measure higher alcohols, acetaldehyde, esters, 4-vinyl guaiacol, and sulfur compounds. Each measurement comprised 5 ml of sample pipetted into a 20 ml glass vial containing 1.75 g NaCl (VWR, 27810.295). 100 µl of 2-heptanol (Sigma-Aldrich, H3003) (internal standard) solution in ethanol (Fisher Chemical, E/0650DF/C17) was added for a final concentration of 2.44 mg/L. Samples were flushed with nitrogen for 10 s, sealed with a silicone septum, stored at −80 °C and analyzed in batches of 20.

The GC was equipped with a DB-WAXetr column (length, 30 m; internal diameter, 0.32 mm; layer thickness, 0.50 µm; Agilent Technologies, Santa Clara, CA, USA) to the FID and an HP-5 column (length, 30 m; internal diameter, 0.25 mm; layer thickness, 0.25 µm; Agilent Technologies, Santa Clara, CA, USA) to the FPD. $N_2$ was used as the carrier gas. Samples were incubated for 20 min at 70 °C in the headspace autosampler (Flow rate, 35 cm/s; Injection volume, 1000 µL; Injection mode, split; Combi PAL autosampler, CTC analytics, Switzerland). The injector, FID and FPD temperatures were kept at 250 °C. The GC oven temperature was first held at 50 °C for 5 min and then allowed to rise to 80 °C at a rate of 5 °C/min, followed by a second ramp of 4 °C/min until 200 °C kept for 3 min and a final ramp of (4 °C/min) until 230 °C for 1 min. Results were analyzed with the GCSolution software version 2.4 (Shimadzu, Kyoto, Japan). The GC was calibrated with a 5% EtOH solution (VWR International) containing the volatiles under study (Supplementary Table S7).

**HS-SPME-GC-MS.** HS-SPME-GC-MS (Shimadzu GCMS-QP-2010 Ultra) was used to measure additional volatile compounds, mainly comprising terpenoids and esters. Samples were analyzed by HS-SPME using a triphase DVB/Carboxen/PDMS 50/30 µm SPME fiber (Supelco Co., Bellefonte, PA, USA) followed by gas chromatography (Thermo Fisher Scientific Trace 1300 series, USA) coupled to a mass spectrometer (Thermo Fisher Scientific ISQ series MS) equipped with a TriPlus RSH autosampler. 5 ml of degassed beer sample was placed in 20 ml vials containing 1.75 g NaCl (VWR, 27810.295). 5 µl internal standard mix was added, containing 2-heptanol (1 g/L) (Sigma-Aldrich, H3003), 4-fluorobenzaldehyde (1 g/L) (Sigma-Aldrich, 128376), 2,3-hexanedione (1 g/L) (Sigma-Aldrich, 144169) and guaiacol (1 g/L) (Sigma-Aldrich, W253200) in ethanol (Fisher Chemical, E/0650DF/C17). Each sample was incubated at 60 °C in the autosampler oven with constant agitation. After 5 min equilibration, the SPME fiber was exposed to the sample headspace for 30 min. The compounds trapped on the fiber were thermally desorbed in the injection port of the chromatograph by heating the fiber for 15 min at 270 °C.

The GC-MS was equipped with a low polarity RXi-5Sil MS column (length, 20 m; internal diameter, 0.18 mm; layer thickness, 0.18 µm; Restek, Bellefonte, PA, USA). Injection was performed in splitless mode at 320 °C, a split flow of 9 ml/min, a purge flow of 5 ml/min and an open valve time of 3 min. To obtain a pulsed injection, a programmed gas flow was used whereby the helium gas flow was set at 2.7 mL/min for 0.1 min, followed by a decrease in flow of 20 ml/min to the normal 0.9 mL/min. The temperature was first held at 30 °C for 3 min and then allowed to rise to 80 °C at a rate of 7 °C/min, followed by a second ramp of 2 °C/min till 125 °C and a final ramp of 8 °C/min with a final temperature of 270 °C.

Mass acquisition range was 33 to 550 amu at a scan rate of 5 scans/s. Electron impact ionization energy was 70 eV. The interface and ion source were kept at 275 °C and 250 °C, respectively. A mix of linear n-alkanes (from C7 to C40, Supelco Co.) was injected into the GC-MS under identical conditions to serve as external retention index markers. Identification and quantification of the compounds were performed using an in-house developed R script as described in Goelen et al. and Reher et al.[87,88] (for package information, see Supplementary Table S8). Briefly, chromatograms were analyzed using AMDIS (v2.71)[89] to separate overlapping peaks and obtain pure compound spectra. The NIST MS Search software (v2.0 g) in combination with the NIST2017, FFNSC3 and Adams4 libraries were used to manually identify the empirical spectra, taking into account the expected retention time. After background subtraction and correcting for retention time shifts between samples run on different days based on alkane ladders, compound elution profiles were extracted and integrated using a file with 284 target compounds of interest, which were either recovered in our identified AMDIS list of spectra or were known to occur in beer. Compound elution profiles were estimated for every peak in every chromatogram over a time-restricted window using weighted non-negative least square analysis after which peak areas were integrated[87,88]. Batch effect correction was performed by normalizing against the most stable internal standard compound, 4-fluorobenzaldehyde. Out of all 284 target compounds that were analyzed, 167 were visually judged to have reliable elution profiles and were used for final analysis.

**Discrete photometric and enzymatic analysis.** Discrete photometric and enzymatic analysis (Thermo Scientific™ Gallery™ Plus Beermaster Discrete Analyzer) was used to measure acetic acid, ammonia, beta-glucan, iso-alpha acids, color, sugars, glycerol, iron, pH, protein, and sulfite. 2 ml of sample volume was used for the analyses. Information regarding the reagents and standard solutions used for analyses and calibrations is included in Supplementary Table S7 and Supplementary Table S9.

**NIR analyses.** NIR analysis (Anton Paar Alcolyzer Beer ME System) was used to measure ethanol. Measurements comprised 50 ml of sample, and a 10% EtOH solution was used for calibration.

**Correlation calculations.** Pairwise Spearman Rank correlations were calculated between all chemical properties.

## Sensory dataset

**Trained panel.** Our trained tasting panel consisted of volunteers who gave prior verbal informed consent. All compounds used for the validation experiment were of food-grade quality. The tasting sessions were approved by the Social and Societal Ethics Committee of the KU Leuven (G-2022-5677-R2(MAR)). All online reviewers agreed to the Terms and Conditions of the RateBeer website.

Sensory analysis was performed according to the American Society of Brewing Chemists (ASBC) Sensory Analysis Methods[90]. 30 volunteers were screened through a series of triangle tests. The sixteen most sensitive and consistent tasters were retained as taste panel members. The resulting panel was diverse in age [22–42, mean: 29], sex [56% male] and nationality [7 different countries]. The panel developed a consensus vocabulary to describe beer aroma, taste and mouthfeel. Panelists were trained to identify and score 50 different attributes, using a 7-point scale to rate attributes' intensity. The scoring sheet is included as Supplementary Data 3. Sensory assessments took place

between 10–12 a.m. The beers were served in black-colored glasses. Per session, between 5 and 12 beers of the same style were tasted at 12 °C to 16 °C. Two reference beers were added to each set and indicated as 'Reference 1 & 2', allowing panel members to calibrate their ratings. Not all panelists were present at every tasting. Scores were scaled by standard deviation and mean-centered per taster. Values are represented as z-scores and clustered by Euclidean distance. Pairwise Spearman correlations were calculated between taste and aroma sensory attributes. Panel consistency was evaluated by repeating samples on different sessions and performing ANOVA to identify differences, using the 'stats' package (v4.2.2) in R (for package information, see Supplementary Table S8).

**Online reviews from a public database.** The 'scrapy' package in Python (v3.6) (for package information, see Supplementary Table S8). was used to collect 232,288 online reviews (mean=922, min=6, max=5343) from RateBeer, an online beer review database. Each review entry comprised 5 numerical scores (appearance, aroma, taste, palate and overall quality) and an optional review text. The total number of reviews per reviewer was collected separately. Numerical scores were scaled and centered per rater, and mean scores were calculated per beer.

For the review texts, the language was estimated using the packages 'langdetect' and 'langid' in Python. Reviews that were classified as English by both packages were kept. Reviewers with fewer than 100 entries overall were discarded. 181,025 reviews from >6000 reviewers from >40 countries remained. Text processing was done using the 'nltk' package in Python. Texts were corrected for slang and misspellings; proper nouns and rare words that are relevant to the beer context were specified and kept as-is ('Chimay','Lambic', etc.). A dictionary of semantically similar sensorial terms, for example 'floral' and 'flower', was created and collapsed together into one term. Words were stemmed and lemmatized to avoid identifying words such as 'acid' and 'acidity' as separate terms. Numbers and punctuation were removed.

Sentences from up to 50 randomly chosen reviews per beer were manually categorized according to the aspect of beer they describe (appearance, aroma, taste, palate, overall quality—not to be confused with the 5 numerical scores described above) or flagged as irrelevant if they contained no useful information. If a beer contained fewer than 50 reviews, all reviews were manually classified. This labeled data set was used to train a model that classified the rest of the sentences for all beers[91]. Sentences describing taste and aroma were extracted, and term frequency–inverse document frequency (TFIDF) was implemented to calculate enrichment scores for sensorial words per beer.

The sex of the tasting subject was not considered when building our sensory database. Instead, results from different panelists were averaged, both for our trained panel (56% male, 44% female) and the RateBeer reviews (70% male, 30% female for RateBeer as a whole).

**Beer price collection and processing.** Beer prices were collected from the following stores: Colruyt, Delhaize, Total Wine, BeerHawk, The Belgian Beer Shop, The Belgian Shop, and Beer of Belgium. Where applicable, prices were converted to Euros and normalized per liter. Spearman correlations were calculated between these prices and mean overall appreciation scores from RateBeer and the taste panel, respectively.

**Correlation calculations.** Pairwise Spearman Rank correlations were calculated between all sensory properties.

**Machine learning models**
**Predictive modeling of sensory profiles from chemical data.** Regression models were constructed to predict (a) trained panel scores for beer flavors and quality from beer chemical profiles and (b) public reviews' appreciation scores from beer chemical profiles. Z-scores were used to represent sensory attributes in both data sets.

Chemical properties with log-normal distributions (Shapiro-Wilk test, $p < 0.05$) were log-transformed. Missing chemical measurements (0.1% of all data) were replaced with mean values per attribute. Observations from 250 beers were randomly separated into a training set (70%, 175 beers) and a test set (30%, 75 beers), stratified per beer style. Chemical measurements (p = 231) were normalized based on the training set average and standard deviation. In total, three linear regression-based models: linear regression with first-order interaction terms (LR), lasso regression with first-order interaction terms (Lasso) and partial least squares regression (PLSR); five decision tree models, Adaboost regressor (ABR), Extra Trees (ET), Gradient Boosting regressor (GBR), Random Forest (RF) and XGBoost regressor (XGBR); one support vector machine model (SVR) and one artificial neural network model (ANN) were trained. The models were implemented using the 'scikit-learn' package (v1.2.2) and 'xgboost' package (v1.7.3) in Python (v3.9.16). Models were trained, and hyperparameters optimized, using five-fold cross-validated grid search with the coefficient of determination ($R^2$) as the evaluation metric. The ANN (scikit-learn's MLPRegressor) was optimized using Bayesian Tree-Structured Parzen Estimator optimization with the 'Optuna' Python package (v3.2.0). Individual models were trained per attribute, and a multi-output model was trained on all attributes simultaneously.

**Model dissection.** GBR was found to outperform other methods, resulting in models with the highest average $R^2$ values in both trained panel and public review data sets. Impurity-based rankings of the most important predictors for each predicted sensorial trait were obtained using the 'scikit-learn' package. To observe the relationships between these chemical properties and their predicted targets, partial dependence plots (PDP) were constructed for the six most important predictors of consumer appreciation[74,75].

The 'SHAP' package in Python (v0.41.0) was implemented to provide an alternative ranking of predictor importance and to visualize the predictors' effects as a function of their concentration[68].

**Validation of causal chemical properties.** To validate the effects of the most important model features on predicted sensory attributes, beers were spiked with the chemical compounds identified by the models and descriptive sensory analyses were carried out according to the American Society of Brewing Chemists (ASBC) protocol[90].

Compound spiking was done 30 min before tasting. Compounds were spiked into fresh beer bottles, that were immediately resealed and inverted three times. Fresh bottles of beer were opened for the same duration, resealed, and inverted thrice, to serve as controls. Pairs of spiked samples and controls were served simultaneously, chilled and in dark glasses as outlined in the *Trained panel* section above. Tasters were instructed to select the glass with the higher flavor intensity for each attribute (directional difference test[92]) and to select the glass they prefer.

The final concentration after spiking was equal to the within-style average, after normalizing by ethanol concentration. This was done to ensure balanced flavor profiles in the final spiked beer. The same methods were applied to improve a non-alcoholic beer. Compounds were the following: ethyl acetate (Merck KGaA, W241415), ethyl hexanoate (Merck KGaA, W243906), isoamyl acetate (Merck KGaA, W205508), phenethyl acetate (Merck KGaA, W285706), ethanol (96%, Colruyt), glycerol (Merck KGaA, W252506), lactic acid (Merck KGaA, 261106).

Significant differences in preference or perceived intensity were determined by performing the two-sided binomial test on each attribute.

**Reporting summary**
Further information on research design is available in the Nature Portfolio Reporting Summary linked to this article.

## Data availability

The data that support the findings of this work are available in the Supplementary Data files and have been deposited to Zenodo under accession code 10653704[93]. The RateBeer scores data are under restricted access, they are not publicly available as they are property of RateBeer (ZX Ventures, USA). Access can be obtained from the authors upon reasonable request and with permission of RateBeer (ZX Ventures, USA). Source data are provided with this paper.

## Code availability

The code for training the machine learning models, analyzing the models, and generating the figures has been deposited to Zenodo under accession code 10653704[93].

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

## Acknowledgements

We thank all lab members for their discussions and thank all tasting panel members for their contributions. Special thanks go out to Dr. Karin Voordeckers for her tremendous help in proofreading and improving the manuscript. M.S. was supported by a Baillet-Latour fellowship, L.C. acknowledges financial support from KU Leuven (C16/17/006), F.A.T. was supported by a PhD fellowship from FWO (1S08821N). Research in the lab of K.J.V. is supported by KU Leuven, FWO, VIB, VLAIO and the Brewing Science Serves Health Fund. Research in the lab of T.W. is supported by FWO (G.0A51.15) and KU Leuven (C16/17/006).

## Author contributions

S.P., M.S. and K.J.V. conceived the experiments. S.P., M.S. and K.J.V. designed the experiments. S.P., M.S., M.R., B.H. and F.A.T. performed the experiments. S.P., M.S., L.C., C.V., L.K., A.B., P.M., L.D., T.W. and K.J.V. contributed analysis ideas. S.P., M.S., L.C., C.V., T.W. and K.J.V. analyzed the data. All authors contributed to writing the manuscript.

## Competing interests

K.J.V. is affiliated with bar.on. The other authors declare no competing interests.
