## [Peer Review File · Nature Communications]

Predicting and improving complex beer flavor through machine learningREVIEWER COMMENTS

Reviewer #1 (Remarks to the Author):

The manuscript by Schreurs et al. reports an in-depth chemical and sensory evaluation (using both professional tasting panels and publicly available data bases) of a large number of commercial beers. The data are used to train ten different machine learning models to predict sensory properties and overall appreciation based on chemical analysis. The predictive power of the models is confirmed by spiking beers with the chemical drivers of consumer appreciation as predicted by the best performing models.

The topic is of significant interest to food and sensory scientists as well as food and beverage-producing industries. As highlighted by the authors, sensory evaluation of complex food products such as wine and beer have remained entirely dependent on expensive and time-consuming sensory panels constituted by product experts or consumer groups, and predictions based on chemical analysis could not be linked to consumer liking or appreciation. Establishing such predictive models has been the target of research for some time, and this manuscript goes some way in bridging the gap between chemical analysis and consumer appreciation.

The manuscript is generally well written, logically structured and was a pleasure to read. The data and figures are of good quality, and the conclusions are mostly justified.

Overall, the manuscript describes some significant scientific achievements.

I do not have any major criticism but several editorial suggestions:

1. The paper would benefit from a more in-depth discussion of the complexity of aroma perception. The authors mention these complexities but do not highlight sufficiently the existing data on such issues. The non-linear perception of varying concentration of specific single compounds, perceived as pleasant at certain concentrations, unpleasant at others, and, more relevant, the interactions between different aroma and/or flavour compounds. Different combinations of two or more compounds, each well-defined individually and described in terms of perception, may result in entirely novel, pleasant or unpleasant perceptions.

2. The authors highlight several limitations of their models and discuss some of the underlying reasons, but the paper would still benefit from a more critical evaluation of the data. There is no doubt that the paper presents an impressive amount of good quality data and, to my knowledge, goes further than any previous attempt in providing predictive modelling of sensory properties based on chemical profiles. However, the success remains heavily dependent on the inclusion of different styles of beer that show clearly described and well-defined sensory properties. It can therefore be argued that many of the findings were predictable and that the models do not reveal fundamentally novel insights. This could be better and more critically contextualised in the discussion.

Specific comments:

1. Line 108 ff lists chemical properties, and mixes general sensory descriptors (bitterness, a complex perception that can be due to many different impact compounds) with specific impact molecules (sugar concentrations). Reword.

2. Line 141 ff: This paragraph is providing some obvious perception vectors linked to specific and very distinctive styles of beer. I am not sure that these observation can be used to claim that the confirm the quality of the measurements. Rather, these descriptors are minimum criteria, and it would have been a serious surprise had they been missed. Reword.

3. I am not sure whether the justification for using "overall appreciation" as a marker for the successful prediction of impact drivers is sufficiently motivated. The authors only shortly explain why they use this assessment criteria, and it can certainly be justified, but a more in-depth discussion of its meaning would be helpful.

Overall, this paper clearly makes a major contribution to the field.

Reviewer #2 (Remarks to the Author):

The manuscript titled "Predicting and improving complex beer flavor through machine learning" measured over 200 chemical properties, conducted panel tasting, and collected consumer reviews from a large online data source for 250 different beers. Machine Learning (ML) was used to

correlate the chemical properties to the panel tasting sensory analysis and the online consumer reviews, separately. From this, feature importance methods determined key chemical properties that drive consumer appreciation. Validation of these findings was conducted through addition of these compounds to beer samples during taste tests. Overall, the study is highly impressive, timely, and novel and I recommend publication following minor revisions.

My main comment relates to the identification of the compounds as drivers of beer flavour and appreciation. I think the following points should be discussed to aid the reader to develop alternative methodologies:

Firstly, given the limitations listed by the authors regarding the online review dataset (price bias, brand bias, conformity to previous ratings, style bias, serving temperature, freshness) and its low correlation to the tasting panel for overall appreciation found in the study, I would think that combining both the panel tasting dataset and the online review dataset would help to overcome the limitations of each dataset (the limitations of the panel tasting dataset being the high variability due to the low sample size). To combine both datasets, the predicted outputs could be averaged between both datasets (e.g., overall appreciation, aroma (averaging malt, hops, and esters for the panel tasting dataset), taste (averaging malt, hops, and esters for the panel tasting dataset), as well as appearance and palate factors are common to both datasets), or the ML models could be trained on both sets of data which would amount to the same effect. The authors state that it has been shown that the consumer review data can "complement or even replace trained tasting panels ... despite biases that are known to occur in such datasets". While it has been validated that the consumer review data enabled identification of compounds that did improve beer appreciation, it has not been compared to compounds identified using other methods (such as tasting panel only or combining datasets) so it is not known whether they should replace taste panels. The authors also state that "since GBR models on our RateBeer dataset showed the best overall performance, we focused on these models". Combining datasets may result in lower performance, however the data would be known to contain less bias. The authors state that "both [feature importance] approaches identified ethyl acetate as the most predictive parameter for beer appreciation". However, it would be more robust to compare different algorithms (e.g., Gradient Boosting Regressors (GBR) and random forests), multiple GBR models, or individual vs combined datasets.

Secondly, given the random element of decision tree training along with the stated limitation of co-correlation of important variables, perhaps multiple GBRs should have been trained and the feature importance's averaged in order to determine the most important compound. Analysis could have also assessed whether these compounds were consistent for each GBR. Alternatively, GBR could have been compared to the next best algorithm (random forests) to see whether the compound recommendations were consistent.

Thirdly, the most important compounds to improve appreciation may differ for each type of beer. Other feature importance methods could account for this. For example, permutation-based methods could be used to alter the chemical property features starting from a specific beer datapoint. Each feature could be altered based on the variance in compounds for that specific beer type to determine the impact on overall appreciation. In this way you can exploit the knowledge of beer type learned by the model. There are likely many more feature importance methods that can achieve something similar.

Other minor comments include: 1) Could the RateBeer data for each beer ID be provided in the supplementary material, at least the RateBeer score, appearance, aroma, taste, palate, and overall appreciation as well as the standard deviation to assess which beers may be more influenced by bias. I did not see it included in the current supplementary material and it would be of great value for other researchers to develop their own methods of using the data you have collected. 2) Line 604: I believe that it should read that the beers were split into a training and test set rather than a training and validation set. This is supported as line 612 goes on to say that five-fold cross-validation was used during model training. 3) Could you please change the format of one of your code documents so that it is searchable and can be copied and pasted.

Reviewer #3 (Remarks to the Author):

This study examined a number of methods to correlate many characteristics of beer attained through taste panels, online public ratings, and chemical analyses. The authors have created a large database of data, and explored correlations created using a variety of methods. The findings were then validated.

The findings are likely to be useful across the entire food industry, however, there are weaknesses in how the findings can be applied. Most of these are already well discussed within the paper however, one that should be expanded upon is how beer style can affect appreciation of specific compounds/properties. This is a weakness of the paper as some of the examples provided in the discussion can be easily explained through style dependency (a positive attribute in one style can be a defect in another). While style distributions ARE discussed and well presented, these do not appear to have been incorporated into the models leading to finding such as lactic acid as a predictor of overall quality. This is likely true of beer styles where lactic acid is expected (sour, fruit, etc.) but unlikely to translate to most other styles. If style is incorporated, the findings of this paper could be much stronger.

Specific corrections:

Graphical Abstract: The term "Belgian beers" should be replaced with "beers for Belgian companies" or equivalent in the graphical abstract to avoid confusion with a perceived style. The differentiation is well explained in the results section, however should be clear from the onset.

Table S1 is a very useful collection of data, however, some of the compounds have potential origin's/flavors and MODs that are not listed. Therefore, the references used to construct this table should be included as part of the table for each entry so that the sources can be evaluated. With the inclusion of these references, the table will become even more useful.

Mouthfeel should replace texture for the sensation experienced in beverages by ethanol and carbon dioxide concentration. This should be applied throughout.

83 Add mashing and aging.

109 Why these compounds in particular? - Please provide sources for why these are significant to beer (they are, but explain why these compounds in particular were chosen and reference accordingly).

135 It should be mentioned that hops are added specifically to inhibit the growth of bacteria that could produce these compounds, not just as a style correlation

148 There are many ways to produce low/non alcohol beverages where this would not be true, this point should be modified or removed.

209 - Was the data provided, or scraped from the website?

221 Define a.o.

249-250 This sentence is misleading. In all disciplines linear relationships are used when appropriate, and non linear relationships are also used when appropriate. Complex nonlinear and dependent relationships have been employed in a variety of FS applications for a long time. This point should be removed.

Table s3 showed some very low performance metrics for SVR, can the authors provide discussion explaining this deviation from the other models?

268 - how was the data split

307 building upon the low appreciation score of non-alcohol beers speculated on at this line, why

was beer style not considered a variable in this analysis. It would improve correlation of factors such as acid/bitterness/alcohol/carbonation which are expected to be high in some styles, but low in others. This would help explain other perceived dilemma such as line 452 – lactic acid would be highly rated in SOUR beer styles, but considered a major defect in most others.

458 - I don't believe the term "absolute prime challenge" is meaningful

REVIEWER COMMENTS to manuscript NCOMMS-23-52289 (Predicting and improving complex beer flavor through machine learning)

Reviewer #1 (Remarks to the Author):

The manuscript by Schreurs et al. reports an in-depth chemical and sensory evaluation (using both professional tasting panels and publicly available data bases) of a large number of commercial beers. The data are used to train ten different machine learning models to predict sensory properties and overall appreciation based on chemical analysis. The predictive power of the models is confirmed by spiking beers with the chemical drivers of consumer appreciation as predicted by the best performing models. The topic is of significant interest to food and sensory scientists as well as food and beverage-producing industries. As highlighted by the authors, sensory evaluation of complex food products such as wine and beer have remained entirely dependent on expensive and time-consuming sensory panels constituted by product experts or consumer groups, and predictions based on chemical analysis could not be linked to consumer liking or appreciation. Establishing such predictive models has been the target of research for some time, and this manuscript goes some way in bridging the gap between chemical analysis and consumer appreciation.

The manuscript is generally well written, logically structured and was a pleasure to read. The data and figures are of good quality, and the conclusions are mostly justified. Overall, the manuscript describes some significant scientific achievements. I do not have any major criticism but several editorial suggestions:

We would like to thank the reviewer for his/her comments and thoughtful suggestions for improvement.

1. The paper would benefit from a more in-depth discussion of the complexity of aroma perception. The authors mention these complexities but do not highlight sufficiently the existing data on such issues. The non-linear perception of varying concentration of specific single compounds, perceived as pleasant at certain concentrations, unpleasant at others, and, more relevant, the interactions between different aroma and/or flavour compounds. Different combinations of two or more compounds, each well-defined individually and described in terms of perception, may result in entirely novel, pleasant or unpleasant perceptions.

We agree and have made the following changes to the introduction (lines 57-61):

“Moreover, sensory analysis is perhaps even more complicated. Flavor perception is highly complex, resulting from hundreds of different molecules interacting at the physiochemical and sensorial level. Sensory perception is often non-linear, characterized by complex and concentration-dependent synergistic and antagonistic effects⁵⁻¹¹ that are further convoluted by the genetics, environment, culture and psychology of consumers¹²⁻¹⁴.”

The new references are:

Meilgaard, M. C. Prediction of flavor differences between beers from their chemical composition. *J. Agric. Food Chem.* 30, 1009–1017 (1982).

Xu, L. *et al.* Widespread receptor-driven modulation in peripheral olfactory coding. *Science* **368**, eaaz5390 (2020).

Kupferschmidt, K. Following the Flavor. *Science* **340**, 808–809 (2013).

Billesbølle, C. B. *et al.* Structural basis of odorant recognition by a human odorant receptor. *Nature* **615**, 742–749 (2023).

Smith, B. Perspective: Complexities of flavour. *Nature* **486**, S6–S6 (2012).

Pfister, P. *et al.* Odorant Receptor Inhibition Is Fundamental to Odor Encoding. *Curr. Biol.* **30**, 2574-2587.e6 (2020).

Ferdenzi, C. *et al.* Variability of Affective Responses to Odors: Culture, Gender, and Olfactory Knowledge. *Chem. Senses* **38**, 175–186 (2013).

2. The authors highlight several limitations of their models and discuss some of the underlying reasons, but the paper would still benefit from a more critical evaluation of the data. There is no doubt that the paper presents an impressive amount of good quality data and, to my knowledge, goes further than any previous attempt in providing predictive modelling of sensory properties based on chemical profiles. However, the success remains heavily dependent on the inclusion of different styles of beer that show clearly described and well-defined sensory properties. It can therefore be argued that many of the findings were predictable and that the models do not reveal fundamentally novel insights. This could be better and more critically contextualised in the discussion.

We thank the reviewer for his/her kind words. We agree that many of the findings are style-dependent, for example the yeast-driven flavors that are common to some Belgian beer styles. However, the set of beers we used for this study is arguably extremely diverse and covers many of the most important beer styles globally. That said, we also acknowledge that some regional or niche styles are missing, and that our models might 1) miss some of the predictors for these styles and 2) might not optimally predict consumer perception for these styles. Colleagues that wish to build upon our study should ensure including samples that cover the entirety of their subject/product/market of interest.

It is true (and reassuring) that several of the compounds identified have been described before as prime drivers of beer aroma, in fact, we believe that this is a good “sanity check” and gives us confidence that our results are correct. However, it is important to point out that we also identify parameters that have not received as much attention as predictors of aroma or appreciation (e.g. protein, methanethiol, isobutanol, caloric value). Moreover, we also show that some other parameters that are often considered as being very specific for only a limited set of beers (e.g. lactic acid) may in fact be interesting and important across all beers. Lastly, a unique feature of our study is that it investigates the influence and importance of all these aroma compounds together, so that the relative importance of each driver becomes more clear, as well as the influence of interactions between aroma drivers.

We have now modified the discussion section to better articulate the style-dependency of the results (lines 482-487):

“Our study is limited to beers from Belgian breweries. Although these beers cover a large portion of the beer styles available globally, some beer styles and consumer patterns may be missing, while other features might be overrepresented. For example, many Belgian ales exhibit yeast-driven flavor profiles, which is reflected in the chemical drivers of appreciation discovered by this study. In future work, expanding the scope to include diverse markets and beer styles could lead to the identification of even more drivers of appreciation and better models for special niche products that were not present in our beer set.”

Specific comments:

1. Line 108 ff lists chemical properties, and mixes general sensory descriptors (bitterness, a complex perception that can be due to many different impact compounds) with specific impact molecules (sugar concentrations). Reword.

We thank the reviewer for pointing out this discrepancy, the modified section now reads (lines 110-112):

“For each beer, we measured 226 different chemical properties, including common brewing parameters such as alcohol content, iso-alpha acids, pH, sugar concentration, and over 200 flavor compounds (see Materials and Methods, Supplemental Table S1).”

2. Line 141 ff: This paragraph is providing some obvious perception vectors linked to specific and very distinctive styles of beer. I am not sure that these observation can be used to claim that the confirm the quality of the measurements. Rather, these descriptors are minimum criteria, and it would have been a serious surprise had they been missed. Reword.

We agree. The section has been adapted and now reads (lines 146-147):

“These observations agree with expectations for key beer styles, and serve as a control for our measurements.”

3. I am not sure whether the justification for using “overall appreciation” as a marker for the successful prediction of impact drivers is sufficiently motivated. The authors only shortly explain why they use this assessment criteria, and it can certainly be justified, but a more in-depth discussion of its meaning would be helpful.

To address this suggestion, we have included the following statement (lines 299-304) to better motivate why we use “overall appreciation”:

“Next, we leveraged our models to infer important contributors to sensory perception and consumer appreciation. Consumer preference is a crucial sensory aspects, because a product that shows low consumer appreciation scores often does not succeed commercially¹⁵. Additionally, the requirement for a large number of representative evaluators makes consumer trials one of the more costly and time-consuming aspects of product development. Hence, a model for predicting chemical drivers of overall appreciation would be a welcome addition to the available toolbox for food development and optimization.”

Overall, this paper clearly makes a major contribution to the field.

Reviewer #2 (Remarks to the Author):

The manuscript titled “Predicting and improving complex beer flavor through machine learning” measured over 200 chemical properties, conducted panel tasting, and collected consumer reviews from a large online data source for 250 different beers. Machine Learning (ML) was used to correlate the chemical properties to the panel tasting sensory analysis and the online consumer reviews, separately. From this, feature importance methods determined key chemical properties that drive consumer appreciation. Validation of these findings was conducted through addition of these compounds to beer samples during taste tests. Overall, the study is highly impressive, timely, and novel and I recommend publication following minor revisions.

We thank the reviewer for his/her insightful comments, especially regarding the machine learning models. We addressed the comments by incorporating the suggested modeling approaches, the results of which are discussed below.

My main comment relates to the identification of the compounds as drivers of beer flavour and appreciation. I think the following points should be discussed to aid the reader to develop alternative methodologies:

Firstly, given the limitations listed by the authors regarding the online review dataset (price bias, brand bias, conformity to previous ratings, style bias, serving temperature, freshness) and its low correlation to the tasting panel for overall appreciation found in the study, I would think that combining both the panel tasting dataset and the online review dataset would help to overcome the limitations of each dataset (the limitations of the panel tasting dataset being the high variability due to the low sample size). To combine both datasets, the predicted outputs could be averaged between both datasets (e.g., overall appreciation, aroma (averaging malt, hops, and esters for the panel tasting dataset), taste (averaging malt, hops, and esters for the panel tasting dataset), as well as appearance and palate factors are common to both datasets), or the ML models could be trained on both sets of data which would amount to the same effect. The authors state that it has been shown that the consumer review data can “complement or even replace trained tasting panels ... despite biases that are known to occur in such datasets”. While it has been validated that the consumer review data enabled identification of compounds that did improve beer appreciation, it has not been compared to compounds identified using other methods (such as tasting panel only or combining datasets) so it is not known whether they should replace taste panels. The authors also state that “since GBR models on our RateBeer dataset showed the best overall performance, we focused on these models”. Combining datasets may result in lower performance, however the data would be known to contain less bias.

We agree that the idea of combining both datasets to level out the biases sounds promising. Perhaps not surprisingly (and as also already predicted by the reviewer), models trained on a combined dataset show a significant drop in performance, from $R^2 = 0.67$ to $R^2 = 0.42$ or $R^2 = 0.26$. The higher value is obtained when we provide the model with a new parameter that identified the datasets. The lower value is obtained when we train the model on the merged dataset without providing an identifier – which would result in the least biased models. In both cases, it seems that the large variation and different bias in both datasets introduces noise, weakening the models’ performances and reliability. In addition, it seems reasonable to assume that both datasets are fundamentally different, with the panel dataset obtained by blind tastings by a trained professional panel, and the RateBeer dataset by (often untrained) consumers

that know exactly which beer they are drinking and are consuming the beer in uncontrolled circumstances. It is well known that such consumer database show biases (eg based on price of the product, marketing, reputation of the product etc). Overall, it seems that the trained panel performed extremely well to identify specific aroma's, but yielded variable results for overall appreciation, which negatively impacts the model's performance. We are aware that a biased model may appear to perform better because it learned these biases. In fact, some of these social parameters could be very interesting to model and separate from patterns that are solely based on chemistry (as is expected to be the case for the data obtained from our trained panel), but this goes beyond the scope of this study and may instead be tackled in follow-up studies.

We have included these new results in the manuscript, see new **Supplementary Fig. S9**.

Supplemental Fig. S9 in our revised manuscript shows the SHAP feature importance for the model trained with a dataset identifier. Interestingly, the 'dataset' identifier is considered most important, suggesting that the model is using this to separate the two datasets and make different predictions based on this information. Many of the most important features of our original model remain in the top-15 (11/15), such as ethyl acetate, ethanol and ethyl octanoate. We do observe that some specific attributes like E-2-nonenal (beer aging marker) and Unfermentables (linked to beer body) are more important compared to the model trained on RateBeer appreciation. This is likely because attributes such as beer aging and beer body are more readily picked up by a trained tasting panel.

A

Supplemental Figure S9: Panel A: Feature importance (MDI and SHAP) of a model trained on both the RateBeer and Trained panel appreciation data. An identifier that encodes the datasets is considered highly important, indicating that the model tries to isolate the datasets before making predictions. Features that are more important for predicting trained panel appreciation, like E-2-nonenal and unfermentables, now appear in the top 15.

We have now added this information in the reworked version of our manuscript (lines 370-379, and **Supplemental Fig. S9**).

*“Next, we investigated if a combination of RateBeer and trained panel data into one consolidated dataset would lead to stronger models, under the hypothesis that such a model would suffer less from bias in the datasets. A GBR model was trained to predict appreciation on the combined dataset. This model underperformed compared to the RateBeer model, both in the native case and when including a dataset identifier ($R^2 = 0.67, 0.26$ and 0.42 respectively). For the latter, the dataset identifier is the most important feature (**Supplemental Fig. S9**), while most of the feature importance remains unchanged, with ethyl acetate and ethanol ranking highest, like in the original model trained only on RateBeer data. It seems that the large variation in the panel dataset continues to introduce noise, weakening the models’ performances and reliability. In addition, it seems reasonable to assume that both datasets are fundamentally different, with the panel dataset obtained by blind tastings by a trained professional panel.”*

The authors state that “both [feature importance] approaches identified ethyl acetate as the most predictive parameter for beer appreciation”. However, it would be more robust to compare different algorithms (e.g., Gradient Boosting Regressors (GBR) and random forests), multiple GBR models, or individual vs combined datasets. Secondly, given the random element of decision tree training along with the stated limitation of co-correlation of important variables, perhaps multiple GBRs should have been trained and the feature importance’s averaged in order to determine the most important compound. Analysis could have also assessed whether these compounds were consistent for each GBR. Alternatively, GBR could have been compared to the next best algorithm (random forests) to see whether the compound recommendations were consistent.

We thank the reviewer for these suggestions. Following the reviewer’s recommendations, we have now run and compared 100 iterations of the best-performing algorithms.

First, we now trained 100 different GBR’s. The results show a remarkable consistency in the top selected features. Compared to the model discussed in our manuscript, the top 13 out of 15 most important features remains the same across the 100 iterations, although some features switch positions (Supplemental Fig. S8, below). Ethyl acetate and ethanol remain the most important features. Regarding performance, the model described in the manuscript reached an R^2 of 0.67 for predicting RateBeer appreciation, and lands among the 25% best-performing models.

Second, when training 100 different Random Forest models, we observe a very similar pattern for the most important features, and here too, ethyl acetate and ethanol remain the most important predictors for appreciation of RateBeer reviewers.

Lastly, when training 100 different iterations of the Extra Trees model, we observe a change in feature importance. Extra Trees is based on the Random Forest architecture, but is specifically designed to be

(even) more random. Most striking is the different shape of the feature importance distribution: due to the increased randomness, the models are no longer systematically selecting ethyl acetate and ethanol as the first splits, which raises the odds of picking other parameters. That said, ethyl acetate and ethanol remain very important for appreciation, ranking second and third. In general, the order of the parameters does change, suggesting co-correlations between parameters. Note that glycerol and ethyl hexanoate, two compounds that we added to our experiments specifically because they co-correlate with ethyl acetate and ethanol, appear in the top 15 features according to the Extra Trees models. Ethyl octanoate, which now ranks number one and was also discovered in our original model, was a good candidate for our follow-up experiments, but unfortunately could not be purchased in food-grade quality.

These results have now been added to the manuscript in the results and discussion section (lines 363 to 369 and 467 to 469):

RESULTS

“To assess the robustness of our best-performing models and model predictions, we performed 100 iterations of the GBR, RF and ET models. In general, all iterations of the models yielded similar performance (Supplemental Fig. S8). Moreover, the main predictors (including the top predictors ethanol and ethyl acetate) remained virtually the same, especially for GBR and RF. For the iterations of the ET model, we did observe more variation in the top predictors, which is likely a consequence of the model’s inherent random architecture in combination with co-correlations between certain predictors. However, even in this case, several of the top predictors (ethanol and ethyl acetate) remain unchanged, although their rank in importance changes (Supplemental Fig. S8).”

DISCUSSION

“As a result, co-correlating variables often have artificially low importance scores, both for impurity and SHAP-based methods, like we observed in the comparison to the more randomized Extra Trees models.”

Supplemental Fig. S8: Results from 100 random iterations of the Gradient Boosting, Random Forest and Extra Trees Regressor. Panel A: Boxplot showing the performance of the different models, measured by the R^2 metric (center line, median; box limits, upper and lower quartiles; whiskers, 1.5x interquartile range; points, outliers). Panel B: Average feature importance, calculated as Mean Decrease in Impurity (MDI). Error bars indicate the standard deviation around the mean. Panel C: Average feature importance, calculated with SHAP. Error bars indicate the standard deviation around the mean.

Figure 3: Feature importance according to the original model, directly copied from the manuscript, for comparison with the results above. Panel A: MDI importance. Panel B: SHAP importance.

Thirdly, the most important compounds to improve appreciation may differ for each type of beer. Other feature importance methods could account for this. For example, permutation-based methods could be used to alter the chemical property features starting from a specific beer datapoint. Each feature could be altered based on the variance in compounds for that specific beer type to determine the impact on overall appreciation. In this way you can exploit the knowledge of beer type learned by the model. There are likely many more feature importance methods that can achieve something similar.

We agree with the reviewer – and with the other reviewers who raised similar comments - that it is interesting to investigate the effect of “beer style” on our predictions and hence check if important compounds for appreciation might differ per beer style.

In response to this suggestion, we have now trained a Gradient Boosting Regression model (GBR or GBM) that had additional information about the beer style. We used one-hot-encoding to create new parameters that encode the style information, and gave these as input to the model. This did not improve the performance of the model ($R^2 = 0.66$ vs $R^2 = 0.67$ for models with and without style information, respectively). Additional results, now added to our reworked manuscript as Supplemental Fig. S9, show that the top 15 most important parameters in this new model remains mostly the same as the ones identified by our original GBR model. Only the ‘Low/no alcohol’ style is now considered very important (likely because it is the least appreciated style in the dataset, discussed below).

Digging deeper at the importance of each style, as shown in Supplemental Table S5, we notice that almost all styles contribute little to the model, ranking last in parameter importance (indicated by red color in the table). Two styles, no/low alcohol and strong ale, do appear to be important, but this is likely because they are the least and most appreciated styles on average. This can be seen in Supplemental Table S6,

which depicts the average RateBeer appreciation score per beer style, with no/low alcohol receiving the lowest scores and Strong Ale receiving the highest. In other words, the model is using the style information to identify the lowest and highest rated beers, which is not useful for our goal of finding chemical drivers of appreciation.

We propose a few explanations for this phenomenon. First, it is important to note that beer style is not a very rigid scientific classification. Beers within one style often differ a lot from each other, including very special “outlier” beers, as well as beers that combine characteristics of multiple styles and, at the same time, lack characteristics that are typically associated with a given style. This further complicates the analysis of style as a factor, and it limits the use of modeling beer aroma within one style. Furthermore, our main goal was to try and identify more general principles that govern aroma perception and appreciation, including compounds that might not be characteristic for a certain beer style but which might in fact help to drive certain aroma’s and/or increase consumer appreciation. We therefore believe that the general models are the best choice.

Second, a lot of styles are likely easily identified based on chemical signatures, for example, lactic and acetic acid are clear proxies for sour beers, ethyl phenolics indicate *Brettanomyces*, relatively high iso-alpha acid values hint at hoppy beers (IPA, Pale Ale). Besides reflecting the beer style, these chemical parameters can also contain additional information, for example iso-alpha acids could also affect the preference for lager beers. Therefore, these parameters may be more informative than style, since they both serve as a proxy for style and contain additional preference information for the other styles.

Lastly, although we have likely gathered one of the largest and most complete datasets, the number of samples per beer style may not be sufficient to detect more subtle effects within or between beer styles (as also noticed by reviewer 3). Given the large number of parameters, our models are still mostly basing their decisions on the values of parameters, and cannot fully grasp the effects of style and interactions¹⁶. Moreover, there is always some inevitable bias in the aroma data that might further limit detecting the influence of beer style. For example, hop aroma is extremely complex, with many of the contributing molecules being very complex and present at low concentrations that are often near or below the detection limit obtained by today’s analyses. As technology improves further, a follow-up study that gathers even more data may therefore be promising to obtain a better insight into the influence of beer style.

The following lines have been added to the results and to the discussion to address the raised concerns (lines 380-390 and 474 – 481), together with **Supplemental Fig. S9**, and **Supplemental Tables S5** and **S6**).

RESULTS

*“Lastly, we evaluated whether beer style identifiers would further enhance the model’s performance. A new GBR model was trained with parameters that explicitly encoded the styles of the samples. This did not improve model performance ($R^2 = 0.66$ with style information vs $R^2 = 0.67$). The most important chemical features are consistent with the model trained without style information (eg. ethanol and ethyl acetate), and with the exception of the most preferred (“strong ale”) and least preferred (“no/low alcohol”) styles, none of the styles were among the most important features (**Supplemental Fig. S9**, **Supplemental Table S5** and **S6**). This is likely due to a combination of style-specific chemical signatures, such as iso-alpha acids and lactic acid, that implicitly convey style information to the original models, as well as the low number of samples belonging to some styles, making it difficult for the model to learn style-specific patterns.*

Moreover, beer styles are not rigorously defined, with some styles overlapping in features and some beers being misattributed to a specific style, all of which leads to more noise in models that use style parameters.”

DISCUSSION

“Expanding our GBR model to include “beer style” as a parameter did not yield additional power or insight. This is likely due to style-specific chemical signatures, such as iso-alpha acids and lactic acid, that implicitly convey style information to the original model, as well as the smaller sample size per style, limiting the power to uncover style-specific patterns. This can be partly attributed to the curse of dimensionality, where the high number of parameters results in the models mainly incorporating single parameter effects, rather than complex interactions such as style-dependent effects¹⁶. A larger number of samples may overcome some of these limitations and offer more insight into style-specific effects. On the other hand, beer style is not a rigid scientific classification, and beers within one style often differ a lot, which further complicates the analysis of style as a model factor.”

B

Supplemental Figure S9: Panel B: Feature importance (MDI and SHAP) of a model trained with a beer style identifier. Besides the inclusion of the “no/low alcohol” style, which is appreciated the least of all the styles on average, the top 15 most important features remain largely the same.

Supplemental Table S5: Feature importance ranks of beer styles in a model trained with style identifiers. Most of the styles are ranked as less important than all chemical compounds (rank 241-253) for predicting RateBeer appreciation. Only two styles (Low/No alcohol and Strong ale) contribute notably to the model's decisions.

Beer style	MDI rank	SHAP rank
Low/No alcohol	11	12
Strong ale	28	17
Blond	173	162
Wheat	206	232
Dubbel	233	236
Flanders old brown	236	229
Faro	237	240
Saison	239	235
Pils/Lager	240	239
Tripel	241	241
Amber	242	243
Hoppy	243	242
Stout/Porter	244	244
West Flanders ale	245	245
Kriek	246	247
Brown	247	246
Fruitbeer	248	248
Lambic	249	249
Brut	250	250
Brett/cofermented	251	251
Christmas	252	252
Scotch	253	253

Supplemental Table S6: Average RateBeer appreciation scores per beer style. The two most important styles according the model (Supplemental Table S5), correspond to the least and most and least appreciated styles, strong ale and low/no alcohol respectively. Please note that our RateBeer scores are negative on average, because each reviewers scores were scaled to all the beers rated by that reviewer. This includes many beers outside of this dataset, which can result in a negative score when averaging over the beers that are included in the study.

Beer style	Average RateBeer appreciation
Low/No alcohol	-2.65
Pils/Lager	-1.68
Fruitbeer	-0.70
Faro	-0.68
Wheat	-0.57
Scotch	-0.44
Amber	-0.37
Blond	-0.35
Flanders old brown	-0.23
Dubbel	-0.17

Hoppy	-0.16
Tripel	-0.15
Stout/Porter	-0.14
West Flanders ale	-0.13
Brut	-0.11
Lambic	-0.09
Saison	-0.08
Christmas	-0.004
Kriek	0.04
Brett/cofermented	0.10
Brown	0.10
Strong ale	0.25

Other minor comments include:

1) Could the RateBeer data for each beer ID be provided in the supplementary material, at least the RateBeer score, appearance, aroma, taste, palate, and overall appreciation as well as the standard deviation to assess which beers may be more influenced by bias. I did not see it included in the current supplementary material and it would be of great value for other researchers to develop their own methods of using the data you have collected.

We understand this request but, following our institute’s legal advice, we cannot include the RateBeer data. While under European law, we can legally use online databases for research purposes, we are not allowed to (re)publish them without prior written consent from the owner, in this case RateBeer, part of ZX Ventures¹⁷. Up until this date, we have not received written consent from RateBeer, and therefore cannot publish the dataset.

2) Line 604: I believe that it should read that the beers were split into a training and test set rather than a training and validation set. This is supported as line 612 goes onto say that five-fold cross-validation was used during model training.

That is indeed a mistake, we have adjusted “validation set” to “test set” (line 675).

3) Could you please change the format of one of your code documents so that it is searchable and can be copied and pasted.

The previous document was a direct PDF export from the Jupyter Notebook to submit it through the journal’s portal, but we agree that this is not user friendly. The general code has been copied to a normal text document that should be easy to open, read and run, which is included in the resubmission (Machine learning models transcript.py). This and all other files are now also made available on Zenodo: <https://doi.org/10.5281/zenodo.10441669>.

Reviewer #3 (Remarks to the Author):

This study examined a number of methods to correlate many characteristics of beer attained through taste panels, online public ratings, and chemical analyses. The authors have created a large database of data, and explored correlations created using a variety of methods. The findings were then validated.

The findings are likely to be useful across the entire food industry, however, there are weaknesses in how the findings can be applied. Most of these are already well discussed within the paper however, one that should be expanded upon is how beer style can affect appreciation of specific compounds/properties. This is a weakness of the paper as some of the examples provided in the discussion can be easily explained through style dependency (a positive attribute in one style can be a defect in another). While style distributions ARE discussed and well presented, these do not appear to have been incorporated into the models leading to finding such as lactic acid as a predictor of overall quality. This is likely true of beer styles where lactic acid is expected (sour, fruit, etc.) but unlikely to translate to most other styles. If style is incorporated, the findings of this paper could be much stronger.

A very similar comment was raised by reviewer #2. Above, we addressed the inclusion of “beer style” and observed no significant improvements. We also propose possible explanations to these results, as well as the related edits made to the manuscript.

Specific corrections:

Graphical Abstract: The term “Belgian beers” should be replaced with “beers for Belgian companies” or equivalent in the graphical abstract to avoid confusion with a perceived style. The differentiation is well explained in the results section, however should be clear from the onset.

We want to thank the reviewer for pointing out this issue. The description of the graphical abstract has been clarified, and now reads (line 34):

“The chemical composition of 250 commercially available beers from Belgian breweries ... “

Table S1 is a very useful collection of data, however, some of the compounds have potential origin’s/flavors and MODs that are not listed. Therefore, the references used to construct this table should be included as part of the table for each entry so that the sources can be evaluated. With the inclusion of these references, the table will become even more useful.

We agree and have now included references per compound in **Supplemental Table S1**.

Mouthfeel should replace texture for the sensation experienced in beverages by ethanol and carbon dioxide concentration. This should be applied throughout.

All mentioning of “texture” has been replaced by “mouthfeel” throughout the entire manuscript.

83 Add mashing and aging.

Mashing and aging are now included in the list, the line now reads (lines 85-86):

"... and biochemical conversions during the brewing process (kilning, mashing, boiling, fermentation, maturation and aging)"

109 Why these compounds in particular? - Please provide sources for why these are significant to beer (they are, but explain why these compounds in particular were chosen and reference accordingly).

A short explanation of the importance of the listed chemicals is now included in the manuscript, and the corresponding references are now listed (lines 110-114):

"For each beer, we measured 226 different chemical properties, including common brewing parameters such as alcohol content, iso-alpha acids, pH, sugar concentration¹⁸, and over 200 flavor compounds (see Materials and Methods, Supplemental Table S1). A large portion (37.2%) are terpenoids arising from hopping, responsible for herbal and fruity flavors^{6,19}. A second major category are yeast metabolites, such as esters and alcohols, that result in fruity and solvent notes¹⁹⁻²¹."

The new references are:

American Society of Brewing Chemists. Beer Methods. (American Society of Brewing Chemists, St. Paul, MN, U.S.A.).

Meilgaard, M. C. Prediction of flavor differences between beers from their chemical composition. J. Agric. Food Chem. 30, 1009–1017 (1982).

Olaniran, A. O., Hiralal, L., Mokoena, M. P. & Pillay, B. Flavour-active volatile compounds in beer: production, regulation and control. J. Inst. Brew. 123, 13–23 (2017).

Verstrepen, K. J. et al. Flavor-active esters: Adding fruitiness to beer. J. Biosci. Bioeng. 96, 110–118 (2003).

Meilgaard, M. C. Flavour chemistry of beer. part I: flavour interaction between principal volatiles. Master Brew. Assoc. Am. Tech. Q. 12, 107–117 (1975).

135 It should be mentioned that hops are added specifically to inhibit the growth of bacteria that could produce these compounds, not just as a style correlation

We thank the reviewer for pointing this out. The manuscript has been modified to better highlight the intent of hops as antimicrobial agents, inhibiting sour bacteria. The lines now read (lines 137-140):

"Similarly, hop-derived iso-alpha acids show a strong anti-correlation with lactic acid and acetic acid, likely reflecting growth inhibition of lactic acid and acetic acid bacteria, or the consequent use of fewer hops in sour beer styles, such as West Flanders ales and Fruit beers, that rely on these bacteria for their distinct flavors²²"

The newly added reference is:

Bossaert, S., Crauwels, S., De Rouck, G. & Lievens, B. The power of sour; A review : Old traditions, new opportunities. *BrewingScience* 72, 78–88 (2019).

148 There are many ways to produce low/non alcohol beverages where this would not be true, this point should be modified or removed.

The non-alcoholic beverages in our dataset are most likely made using cold-contact fermentation or (early generation) thin film evaporation. In all cases, we observe lower concentrations of esters and glycerol. The line has been modified to clarify that the statement reflects the beers in our dataset, and is not a general rule, it now reads (lines 153-157):

“This is in line with the production process for most of the low/no alcohol beers in our dataset, which are produced through limiting fermentation or by stripping away alcohol via evaporation or dialysis, with both methods having the unintended side-effect of reducing the amount of flavor compounds in the final beer.”

209 – Was the data provided, or scraped from the website?

The dataset was scraped from the RateBeer website. Use of the data is legal under the EU law, which makes an exemption on the copyright of online databases, if used for research purposes¹⁷.

221 Define a.o.

“a.o.” has now been written out in full, the line now reads (lines 227-228):

“... and can further contribute to (among others, appreciation) differences between the two categories of tasters.”

If the reviewer instead meant that we need to write out the “others” in full, please let us know and we will make the required changes.

249-250 This sentence is misleading. In all disciplines linear relationships are used when appropriate, and non linear relationships are also used when appropriate. Complex nonlinear and dependent relationships have been employed in a variety of FS applications for a long time. This point should be removed.

We agree with the reviewer’s comment that appropriate tools have existed and are in use for quite some time. However, even when looking at recent literature, we still notice a lot of less-ideal use of statistical tools, including linear methods ((M)ANOVA, PCA, MFA), some of which also do not account for interactions. We want to stress that we do not wish to discredit these studies; in many cases, a linear approximation works fine and leads to the same conclusions. Often, the experimental work is performed very robustly and we have faith in the results and findings of our colleagues in the field. However, we strongly encourage the use of the most appropriate tools for dealing with complex datasets like these, tools that do not make specific assumptions about the underlying data structures.

We do agree that the phrasing “frequently” and “do not suffice” is too strong in our original statement, and therefore adjusted the lines to now read (lines 255-257):

“Given the complexity of beer flavor, basic statistical tools such as correlations or linear regression, may not always be the most suitable for making accurate predictions”

Table s3 showed some very low performance metrics for SVR, can the authors provide discussion explaining this deviation from the other models?

The results shown in **Table 1** in the manuscript were obtained with the MultiOutputRegressor function in python. In short, this function allows models (that natively only handle one output variable) to be trained in parallel for multiple outputs, with the same model settings (=hyperparameters) for all. The overall model performance is then evaluated by taking the mean or median performance on each individual output.

If we instead look at individual models trained per attribute, shown in **Supplemental Table S4**, we observe strikingly poor scores for the “Palate” score in RateBeer. It is likely that the Palate attribute in the RateBeer dataset is negatively skewing the general performance of the SVM. For all the other attributes, we observe decent results of the SVM, although still inferior to the Tree-based models and the Lasso regression model.

The following lines (294-296) were added to clarify this:

“The SVR shows intermediate performance, mostly due to the weak predictions of specific attributes that lower the overall performance (Supplemental Table S4).”

Supplemental Table S4: Performance (R^2) of models trained on individual attributes of the RateBeer dataset. The highlight shows that for the Support Vector Machine (SVM), the “Palate” descriptor has a drastically negative score. For the other attributes, the SVM performs reasonably well, though it remains inferior to the tree-based models.

Attribute	ABR	ET	GBR	Lasso	LM	ANN	PLSR	RF	SVM	XGBR
Ratebeer score	0.6	0.62	0.67	0.62	-11.41	0.7	0.62	0.62	0.58	0.65
Aroma appreciation	0.64	0.64	0.71	0.63	-9.19	0.69	0.63	0.65	0.64	0.68
Appearance	0.59	0.63	0.68	0.61	-10.3	0.61	0.03	0.62	0.61	0.66
Taste appreciation	0.6	0.61	0.71	0.66	-11.69	0.62	0.63	0.64	0.58	0.67
Palate	0.6	0.62	0.64	0.63	-12.37	0.51	0.36	0.65	-4.42	0.65
Overall appreciation	0.6	0.6	0.67	0.6	-11.29	0.61	0.62	0.61	0.56	0.69

268 – how was the data split

The data split was random, but stratified over beer style (*i.e.* keeping a balanced representation of beer styles in both datasets). This was done with the train_test_split function from the sklearn Python package. The manuscript result section has been edited to further clarify this, it now reads (lines 274-275):

“... the dataset was randomly split into a training and test set, stratified by beer style”

307 building upon the low appreciation score of non-alcohol beers speculated on at this line, why was beer style not considered a variable in this analysis. It would improve correlation of factors such as acid/bitterness/alcohol/carbonation which are expected to be high in some styles, but low in others. This would help explain other perceived dilemma such as line 452 – lactic acid would be highly rated in SOUR beer styles, but considered a major defect in most others.

This is also addressed at the start of this document, under the header *“General remarks on the influence and importance of Style”*.

458 - I don't believe the term “absolute prime challenge” is meaningful

The phrasing of this line has been modified, it now reads (line 514):

“... which is generally considered one of the key challenges for future beer production.”

REFERENCES

1. Seitz, H. K. & Stickel, F. Molecular mechanisms of alcohol-mediated carcinogenesis. *Nat. Rev. Cancer* **7**, (2007).
2. Voordeckers, K. *et al.* Ethanol exposure increases mutation rate through error-prone polymerases. *Nat. Commun.* **11**, 3664 (2020).
3. Koubaa, Y. & Eleuch, A. Gender effects on odor-induced taste enhancement and subsequent food consumption. *J. Consum. Mark.* **37**, 511–519 (2020).
4. Duffy, V. B. & Bartoshuk, L. M. Food Acceptance and Genetic Variation in Taste. *J. Am. Diet. Assoc.* **100**, 647–655 (2000).
5. Shepherd, G. M. Smell images and the flavour system in the human brain. *Nature* **444**, 316–321 (2006).
6. Meilgaard, M. C. Prediction of flavor differences between beers from their chemical composition. *J. Agric. Food Chem.* **30**, 1009–1017 (1982).
7. Xu, L. *et al.* Widespread receptor-driven modulation in peripheral olfactory coding. *Science* **368**, (2020).
8. Kupferschmidt, K. Following the Flavor. *Science* **340**, 808–809 (2013).
9. Billesbølle, C. B. *et al.* Structural basis of odorant recognition by a human odorant receptor. *Nature* **615**, 742–749 (2023).
10. Smith, B. Perspective: Complexities of flavour. *Nature* **486**, S6–S6 (2012).
11. Pfister, P. *et al.* Odorant Receptor Inhibition Is Fundamental to Odor Encoding. *Curr. Biol.* **30**, 2574–2587 (2020).
12. Moskowitz, H. W., Kumaraiah, V., Sharma, K. N., Jacobs, H. L. & Sharma, S. D. Cross-cultural differences in simple taste preferences. *Science* **190**, 1217–1218 (1975).
13. Eriksson, N. *et al.* A genetic variant near olfactory receptor genes influences cilantro preference. *Flavour* **1**, 22 (2012).
14. Ferdenzi, C. *et al.* Variability of Affective Responses to Odors: Culture, Gender, and Olfactory Knowledge. *Chem. Senses* **38**, 175–186 (2013).
15. Lawless, H. T. & Heymann, H. *Sensory Evaluation of Food: Principles and Practices*. (Springer, 2010). doi:10.1007/978-1-4419-6488-5.
16. Gries, S. Th. *Statistics for Linguistics with R*. (Walter de Gruyter GmbH, 2021).
17. Directive 96/9/EC of the European Parliament and of the Council of 11 March 1996 on the legal protection of databases. *OJ L* vol. 077 (1996).
18. American Society of Brewing Chemists. *ASBC Beer Methods*.

19. Olaniran, A. O., Hiralal, L., Mokoena, M. P. & Pillay, B. Flavour-active volatile compounds in beer: production, regulation and control. *J. Inst. Brew.* **123**, 13–23 (2017).
20. Verstrepen, K. J. *et al.* Flavor-active esters: Adding fruitiness to beer. *J. Biosci. Bioeng.* **96**, 110–118 (2003).
21. Meilgaard, M. C. Flavour chemistry of beer. part I: flavour interaction between principal volatiles. *Master Brew. Assoc. Am. Tech. Q.* **12**, 107–117 (1975).
22. Bossaert, S., Crauwels, S., De Rouck, G. & Lievens, B. The power of sour - A review : Old traditions, new opportunities. *BrewingScience* **72**, 78–88 (2019).

REVIEWERS' COMMENTS

Reviewer #1 (Remarks to the Author):

The authors have addressed all comments by the reviewers in a more than satisfactory manner. I have no additional comments and recommend publication.

Reviewer #2 (Remarks to the Author):

All my comments have been addressed

Reviewer #3 (Remarks to the Author):

The authors have addressed my concerns thoroughly, I believe the revised manuscript is suitable for publication.